# Aberrant expression of CITED2 promotes prostate cancer metastasis by activating the nucleolin-AKT pathway

Seung-Hyun Shin[1,2,3], Ga Young Lee[1,2,3], Mingyu Lee[1,2,3], Jengmin Kang[1,2,3], Hyun-Woo Shin [1,2,3], Yang-Sook Chun[1,3] & Jong-Wan Park [1,2,3]

Despite many efforts to develop hormone therapy and chemotherapy, no effective strategy to suppress prostate cancer metastasis has been established because the metastasis is not well understood. We here investigate a role of CBP/p300-interacting transactivator with E/D-rich carboxy-terminal domain-2 (CITED2) in prostate cancer metastasis. CITED2 is highly expressed in metastatic prostate cancer, and its expression is correlated with poor survival. The *CITED2* gene is highly activated by ETS-related gene that is overexpressed due to chromosomal translocation. CITED2 acts as a molecular chaperone to guide PRMT5 and p300 to nucleolin, thereby activating nucleolin. Informatics and experimental data suggest that the CITED2–nucleolin axis is involved in prostate cancer metastasis. This axis stimulates cell migration through the epithelial–mesenchymal transition and promotes cancer metastasis in a xenograft mouse model. Our results suggest that CITED2 plays a metastasis-promoting role in prostate cancer and thus could be a target for preventing prostate cancer metastasis.

[1] Department of Biomedical Science, BK21-plus Education Program, Seoul National University College of Medicine, Seoul, Korea. [2] Department of Pharmacology, Seoul National University College of Medicine, Seoul, Korea. [3] Cancer Research Institute and Ischemic/Hypoxic Disease Institute, Seoul National University College of Medicine, Seoul, Korea. Correspondence and requests for materials should be addressed to J.-W.P. (email: parkjw@snu.ac.kr)

Prostate cancer is the most frequently diagnosed cancer and the second leading cause of cancer-related death among males. Despite many efforts to develop hormone therapy and chemotherapy, the prognoses of patients with advanced prostate cancer remains poor, because these treatments cannot control cancer metastasis[1,2]. One of the most distinct features of prostate cancer is that more than half of the patients display gene fusion between androgen-responsive gene TMPRSS2 (trans-membrane protease, serine 2) and ETS (erythroblast transformation-specific) transcription factor genes such as ERG (ETS-related gene) and ETV1[3]. TMPRSS2–ERG fusion is reported to promote cancer progression[4,5], but the downstream mechanism is not clearly known.

CBP/p300-interacting transactivator with E/D-rich carboxy-terminal domain-2 (CITED2, also known as MRG1 and p35srj) is a transcriptional coregulator together with the transcriptional coactivator p300/CBP. Depending on its target gene, it functions as a positive or negative regulator of gene expression. For example, CITED2 acts as a coactivator of activator protein 2 (AP-2) transcription factors by recruiting p300/CBP to AP-2 target genes[6]. In contrast, CITED2 inhibits hypoxia-induced gene expression by preventing p300/CBP recruitment to the hypoxia-inducible factor-1α[7]. CITED2 interacts with other components besides the aforementioned proteins. CITED2 expression is induced by hypoxia, lipopolysaccharides, growth factors, and proinflammatory cytokines[8]. CITED2 also plays essential roles in embryonic stem cell differentiation[9] and development of diverse organs, including liver[10], lung[11], heart[12], and lens[13]. Furthermore, adult hematopoietic stem cell (HSC) functions are maintained by CITED2 via lnk4a/Arf and Trp53[14], and acute myeloid leukemia critically requires CITED2 expression[15]. However, only a few investigations have been conducted on the role of CITED2 in tumor development during the last decade. CITED2 was reported to promote tumorigenesis of Rat1 cells[8] and growth of lung cancer cells[16]. However, CITED2 inhibited proliferation of colon cancer cells[17], and low expression of CITED2 was associated with a poor prognosis in breast cancer[18]. In particular, CITED2 is suspected to be extensively involved in prostate cancer, since its expression is induced by an ETS family member ELK1[19], which has been reported to recruit AR to activate growth signaling in prostate cancer cells[20]. In this study, we performed co-immunoprecipitation and shotgun proteomics to discover a CITED2-interacting protein, and identified nucleolin (NCL). NCL is an RNA-binding nulceolar protein which has been reported to stimulate cancer progression and metastasis[21–23], although the exact underlying mechanism has not been determined.

NCL is widely known to regulate ribosomal RNA (rRNA) transcription of the engrafting complex of pre-ribosomes. NCL binds to non-transcribed spacers of recombinant DNA transcription initiation sites or interacts with histone-1 to induce de-condensation of chromatin structures[24,25]. NCL also forms the pre-rRNA processing complex by recruiting U3 small nucleolar RNA[26,27]. Moreover, NCL promotes translation of target messenger RNAs (mRNAs) by binding to their G-rich mRNA coding regions to facilitate polysome formation on transcripts[28]. NCL consists of three functional domains: the N-terminal domain composed of highly acidic regions intermixed with basic regions, the RNA-binding domain, and the glycine- and arginine-rich domain. NCL is post-translationally modified by casein kinase 2 and p43$^{cdc2}$, which phosphorylate NCL at serine residues within the acidic regions[29] and at threonine residues within the basic regions, respectively[30]. These phosphorylation events of NCL are regulated throughout the cell cycle. Notably, P300-mediated acetylation[31] and PRMT5-mediated methylation[32] of NCL have also been reported, but no studies have been conducted on the oncogenic functional changes induced by these post-translational modifications of NCL.

In the present study, we found that CITED2 was highly expressed in metastatic prostate cancer because of TMPRSS2–ERG gene fusion, which promoted metastasis by activating NCL at the post-translational level. We also propose that the CITED2–NCL signaling pathway is a potential target for treating prostate cancer metastasis.

## Results

**CITED2 is highly expressed in metastatic prostate cancer.** We examined CITED2 expression in 28 different types of cancer using The Cancer Genome Atlas (TCGA) database and found relatively high CITED2 mRNA levels in thyroid, kidney, ovarian, lung, prostate, breast, and lung cancers (Fig. 1a). We next compared CITED2 levels between normal and cancer tissues using the Genomic Spatial Event (GSE) database. Of six types of cancers evaluated, CITED2 was elevated only in prostate cancer compared with normal tissue (Fig. 1b). Prostate cancer patients from the TCGA database were categorized into CITED2_low and CITED2_high groups with respect to the median CITED2 expression value. Overall survival was lower in the CITED2_high group than in the CITED2_low group (Fig. 1c). In thyroid, ovarian, lung, and breast cancers, CITED2 expression was not correlated with overall survival (Supplementary Figure 1a). In renal cell carcinoma (RCC), the CITED2_high group showed a longer survival than that of the CITED2_low group. To examine the involvement of CITED2 in prostate cancer progression, we determined CITED2 expression in primary tumor and metastatic tumor tissues and found that CITED2 expression was increased in metastatic tumors (Fig. 1d). CITED2 expression in other types of cancers was not significantly increased with tumor stage (Supplementary Figure 1b). To evaluate CITED2 expression at the protein level, immunohistochemistry (IHC) using an anti-CITED2 antibody was performed in human prostate cancer tissues, which were categorized according to their Gleason score. Increased CITED2 protein levels were associated with an increasing Gleason score (Fig. 1e). When the prostate cancer tissues were divided into the CITED2_high and CITED2_low groups, a lower tumor-free survival was evident in CITED2_high compared with CITED2_low (Fig. 1f) group. CITED2 expression might correlate with poor prognosis in prostate cancer patients.

**ERG increases CITED2 expression at transcription level in prostate cancer.** The ETS genes are fused to the promoters of the androgen receptor target genes, leading to their high expression in prostate cancer cells[3,33]. Because ELK1 in the ETS family has been reported to transactivate the CITED2 gene[19,34], we examined which member in the ETS family is responsible for CITED2 gene activation in prostate cancer. We compared the mRNA levels of ETS members between normal prostate and prostate cancer tissues using the GSE6919 prostate cancer data set (Supplementary Figure 2a). Among those mRNAs, the ERG level increased to the greatest extent in cancer tissues (Fig. 2a). The rate of gene fusion was highest to ERG among the ETS members according to the TCGA mutation sequence data (Supplementary Figure 2b). Immunoblotting analysis in various prostate cancer cell lines showed an apparent correlation between ERG and CITED2 expressions (Supplementary Figure 2c). Of the examined cell lines, VCaP harboring the TMPRSS2–ERG gene fusion expressed both ERG and CITED2 to the highest levels. When ERG was knocked down using three different small interfering RNAs (siRNAs), the CITED2 protein and mRNA expression were both significantly downregulated in three cell lines, indicating ERG-dependent expression of CITED2 (Fig. 2b, c). In prostate

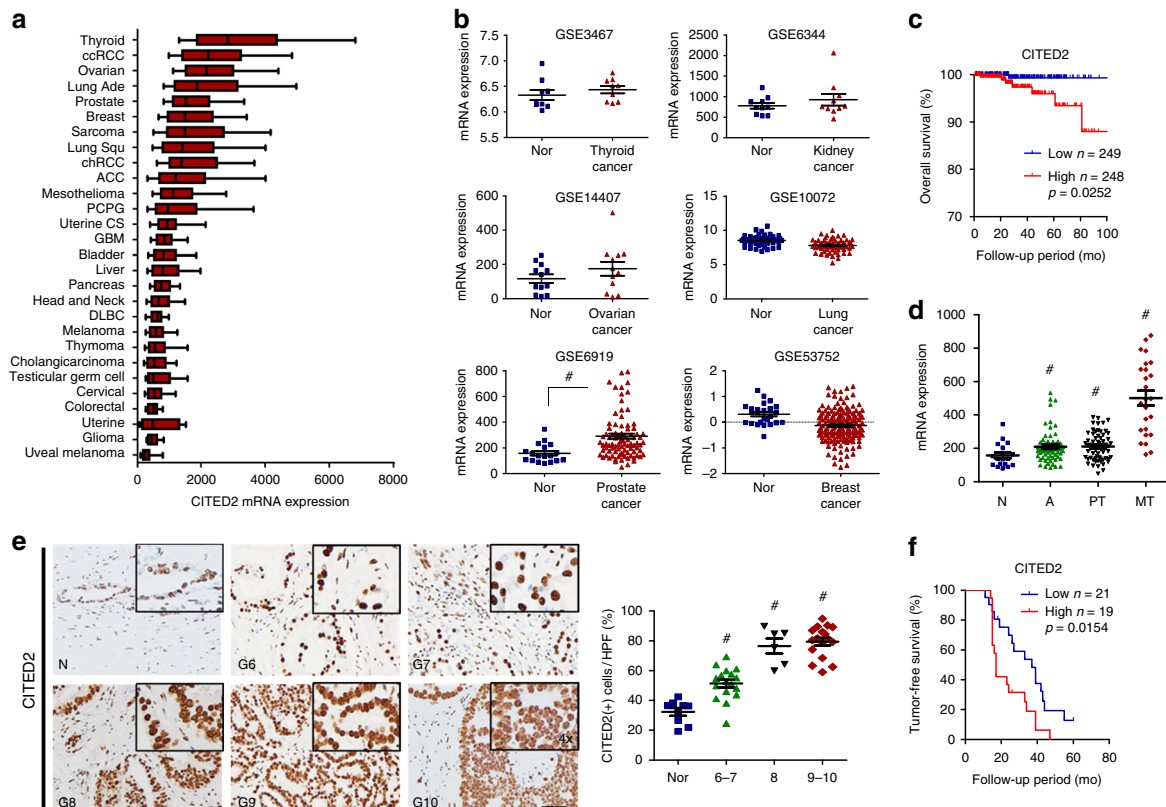

**Fig. 1** CITED2 overexpression is associated with poor prognosis in prostate cancer. **a** CITED2 mRNA expression in 28 different types of cancer as determined by RNA-Seq V2 RSEM. Box plots indicate the distribution of values and the middle thick line in the box shows mean value and whiskers represent 10–90th percentile. Results were obtained from The Cancer Genome Atlas (TCGA) cancer provisional data sets based on TCGA Research Network (http://cancergenome.nih.gov). The abbreviations and the number of patients are described in the Methods section. **b** CITED2 mRNA levels in normal (blue squares) and cancer tissues (red triangles) of thyroid, kidney, ovarian, lung, prostate, and breast (GSE data sets). **c** Kaplan–Meier overall survival analysis of prostate cancer patients on TCGA. *P* value was calculated by log-rank test. **d** CITED2 mRNA levels in human prostate cancer tissues of data set GES6919. N normal prostate tissue (blue squares), A normal prostate tissue adjacent tumor tissue (green triangles), PT prostate primary tumor tissue (black inverted triangles), MT metastatic prostate tumor tissue (red diamond). **e** Representative images of human prostate adenocarcinoma tissues immunostained with anti-CITED2 antibody. Abbreviations at the left bottoms of images: N normal prostate tissue (blue squares), G6–10 (6–7, green triangles; 8, black inverted triangles; 9–10, red diamond) Gleason scores 6–10 of prostate cancer (left panel). The immunostaining scores were calculated by counting stained cells and presented as dot plots (right panel). The scale bar represents 50 μm. **f** Kaplan–Meier tumor-free survival analysis of prostate cancer patients. *P* value was calculated by log-rank test. The horizontal lines in all dot plots represent the means ± SE and $^{\#}P < 0.05$ versus the normal tissue group by Mann–Whitney statistical analysis

cancer cells harboring the TMPRSS2–ERG fusion, ERG expression is known to be highly induced by testosterone. As expected, testosterone robustly induced ERG expression in VCaP cells, where CITED2 expression was subsequently increased. Such effects of testosterone were not observed in DU145 cells without the TMPRSS2–ERG fusion (Fig. 2d). To examine the ERG binding to the CITED2 promoter, we performed chromatin immunoprecipitation and quantitative PCR analyses. Among the three regions within the promoter, the second region was identified as an ERG-binding site (Fig. 2e). We then constructed a luciferase reporter plasmid containing the CITED2 promoter. Compared to PC3 and DU145 with a lower level of ERG, three prostate cancer cell lines with high ERG expression had greater luciferase activity (Supplementary Figure 2d). In these cell lines, CITED2 promoter activity was diminished by knocking down ERG or by mutating the putative ERG-binding motif (Fig. 2f). Next, we performed IHC to characterize ERG expression in prostate cancer tissues. The ERG level in prostate cancer increased with the Gleason score (Fig. 2g). Tumor-free survival in the ERG_high group was significantly lower compared with the ERG_low group (Supplementary Figure 2e). Pearson's correlation

analyses showed that CITED2 expression was positively correlated with ERG expression (Fig. 2h). Furthermore, we performed PCR using DNAs extracted from prostate cancer tissues and detected *TMPRSS2–ERG* gene fusion in 15 of 49 prostate cancers (Fig. 2i). A chi-square test revealed that *TMPRSS2–ERG* gene fusion is associated with a high Gleason score (Supplementary Figure 2f). ERG and CITED2 overexpression in the *TMPRSS2–ERG* gene fusion samples (Fig. 2j) further support the ERG-driven overexpression of CITED2 in prostate cancer cells.

**CITED2 binds to a multimeric complex consisting of NCL, p300, and PRMT5**. To identify the CITED2-interacting proteins, we pulled down the FLAG/SBP-tagged CITED2 construct that was overexpressed in HEK293T cells using anti-FLAG or strep-tavidin affinity beads, and analyzed the co-purified proteins using liquid chromatography-tandem mass spectrometry (LC-MS/MS). Proteins pulled down commonly by anti-FLAG antibody (red) and streptavidin (blue) are listed in Supplementary Data 1. In addition to p300 and CBP, the PRMT5 complex subunits PRMT5, WDR77, and RIOK1, as well as NCL were co-purified with CITED2 (Fig. 3a). The interaction between PRMT5 and

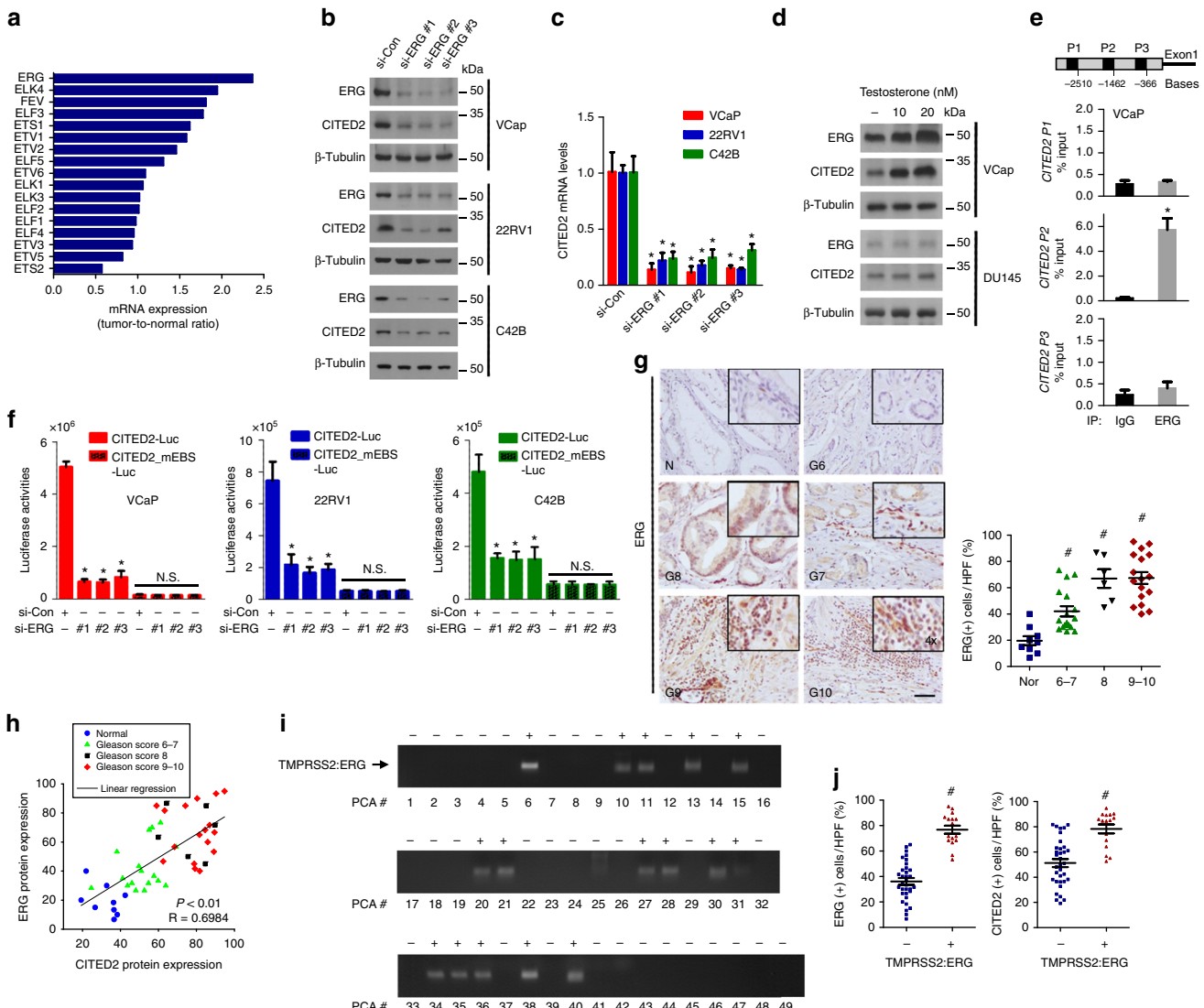

**Fig. 2** ERG gene activation is responsible for CITED2 overexpression in prostate cancer. **a** The mRNA levels of the ETS family members in GSE6919 data set. Each bar represents the ratio of tumor level to normal level. **b** Indicated cells were transfected with non-targeting siRNA (si-Con) or three different ERG-targeting siRNAs (si-ERG #1, #2 and #3) at an 80 nM concentration. Cell lysates were immunoblotted using the indicated antibodies. **c** RT-qPCR was performed to analyze CITED2 mRNA levels in prostate cancer cells transfected with the indicated siRNAs. Each bar represents the mean + SD ($n = 3$). **d** VCaP and DU145 cells were treated with testosterone, and the cell lysates were immunoblotted with anti-ERG, anti-CITED2, or anti-β-tubulin antibody. **e** The ERG binding to three *CITED2* promotor regions (P1, P2 and P3) was detected by ChIP-qPCR using anti-ERG antiserum or non-immunized serum (IgG). The co-precipitated *CITED2* promotor regions was quantified using RT-qPCR. Results (the mean ± SD, $n = 3$) were represented as percentages of IP signal/input signal (% input). *$P < 0.05$ versus the IgG control group. **f** The indicated siRNAs were cotransfected with the CITED2-luciferase or CITED2_mEBS (mutated ERG-binding sequence, patterned) reporter plasmid. Luciferase activities (mean + SD, $n = 3$) were measured by a luminometer. **g** Representative images of human prostate adenocarcinoma tissues immunostained with anti-ERG antibody. Abbreviations at the left bottoms of images: N normal prostate tissue (blue squares), G6–10 (6–7, green triangles; 8, black inverted triangles; 9–10, red diamond) Gleason scores 6–10 of prostate cancer (left panel). The immunostaining scores were calculated by counting stained cells and presented as dot plots (right panel). The scale bar represents 50 μm. **h** Protein expression scatter diagrams of ERG versus CITED2. Linear regression and correlation of ERG versus CITED2. $R^2$ means coefficient of determination calculated by Pearson's correlation. **i** DNAs extracted from prostate cancer tissues were subjected to PCR using a forward primer binding to the *TMPRSS2* promoter and a reverse primer binding to the *ERG* gene. The PCR products of the TMPRSS2–ERG fusion gene (+, positive; −, negative) are indicated as an arrow. **j** The immunostaining scores were calculated by counting stained cells. Blue and red symbols indicate ERG (right panel) and CITED2 (left panel) levels in the TMPRSS2–ERG fusion negative (blue squares) and positive (red triangles) group. The horizontal lines in all dot plots represent the means ± SE and #$P < 0.05$ versus the normal group by Mann–Whitney statistical analysis; *$P < 0.05$ versus the si-Con group by Student's *t*-test; N.S. 'not significantly different' among the groups

CITED2 was verified by immunoprecipitation and immunoblotting using HEK293T cells coexpressing MYC-PRMT5 and Flag/SBP-CITED2 (Supplementary Figure 3a). We analyzed the interactions among endogenous PRMT5, NCL, WDR77, RIOK1, and CITED2 and found that both CITED2 and PRMT5 interacted with NCL, WDR77, and RIOK1 (Supplementary Figure 3b). These protein interactions were confirmed in the immunoprecipitates using NCL, WDR77, or RIOK1 antibodies (Supplementary Figure 3c). As previously shown in HEK293T-cells, these interactions were also identified in all prostate cancer

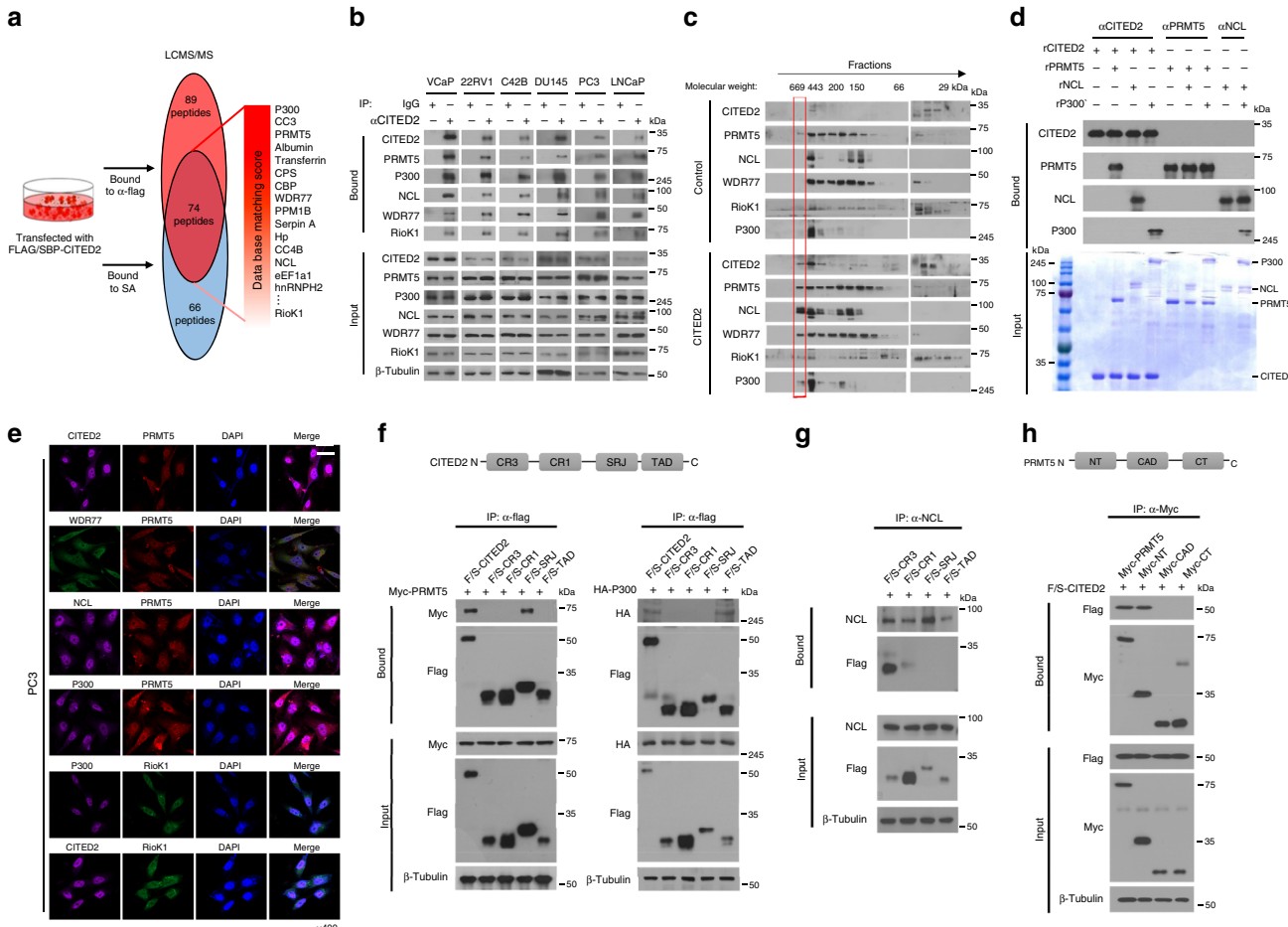

**Fig. 3** CITED2 forms a multimeric complex with nucleolin, p300, and PRMT5 subunits. **a** HEK293T cells were transfected with the Flag/SBP-CITED2 plasmid. CITED2-interacting proteins were precipitated using anti-Flag tag antibody or streptavidin, and identified by LC-MS/MS analyses. The proteins identified commonly in two precipitates are listed. **b** Prostate cancer cell lysates were immunoprecipitated with anti-CITED2 antibody or IgG, and the precipitates were immunoblotted with the indicated antibodies. **c** HEK293T cells were transfected with pcDNA or CITED2, and the cell lysates were subjected to FPLC. The FPLC elutes were immunoblotted with the indicated antibodies. The red box indicates a bigger-sized complex (600 to 700 kDa) that is formed by CITED2 overexpression. **d** In vitro binding analysis. Recombinant protein CITED2, PRMT5, NCL, and P300 were put together in a test tube. Proteins in tube were immunoprecipitated with indicated antibodies, and the precipitates were immunoblotted. Input levels were verified by electrophoresis and Coomassie staining. **e** Representative immunofluorescence images. PC3 cells were grown on coverslips, fixed with methanol, and stained with the indicated antibodies. All samples were stained with DAPI to visualize nuclei. The scale bar represents 20 μm. **f** The Flag/SBP-CITED2 constructs are shown in the top panel. HEK293T cells were cotransfected with one of the CITED2 constructs and Myc-PRMT5, HA-P300, and Flag/SBP-peptides were immunoprecipitated with anti-Flag and Myc-PRMT5 and HA-P300 were detected by western blotting. **g** HEK293T cells were transfected with the CITED2 constructs, and cell lysates were immunoprecipitated with anti-NCL and Flag/SBP-CITED2 peptides were detected by western blotting. **h** The Myc-PRMT5 constructs are shown in the top panel. HEK293T cells were cotransfected with one of the PRMT5 construct and Flag/SBP-CITED2, and Myc-peptides were immunoprecipitated with anti-Myc and Flag/SBP-CITED2 were detected by western blotting. All experiments were carried out at three distinct samples

cell lines examined (Fig. 3b). To verify the presence of this multimeric complex, we separated intracellular proteins using fast protein LC. Standard protein markers were used to determine the molecular weight of each fraction (Supplementary Figure 3d). CITED2, PRMT5, NCL, WDR77, Riok1, and p300 were detected in the ~500 kDa fraction. Importantly, when CITED2 was over-expressed, the complex was shifted to ~700 kDa (Fig. 3c), which suggested that CITED2 plays a role in attracting proteins to the complex. To examine if CITED2, NCL, P300, and PRMT5 are directly associated, an in vitro binding assay was conducted using recombinant proteins. CITED2 directly interacted with NCL, P300, and PRMT5, while PRMT5 did not bind to P300 and NCL. NCL and P300 were also bound directly (Fig. 3d). Next, we performed immunofluorescent staining to determine the

subcellular location of the proteins. The subunits in the complex were co-localized mainly in the nuclei of PC3 (Fig. 3e) or HEK293T cells (Supplementary Figure 3e). The interaction between PRMT5 and CITED2 was further characterized by immunoprecipitation of the domain peptides of CITED2 and PRMT5. p300, PRMT5, and NCL were identified to bind to the transactivation domain, the serine/glycine-rich junction (SRJ), and the cysteine/arginine-rich domain 3 (CR3) of CITED2, respectively (Fig. 3f, g). In addition, CITED2 interacted with the N terminus of PRMT5 (Fig. 3h). Since CITED2 provides different binding sites for p300, PRMT5, and NCL, these proteins could form a stable complex in a noncompetitive manner. Therefore, we hypothesized that CITED2 acts as an essential binder to make the multimeric complex.

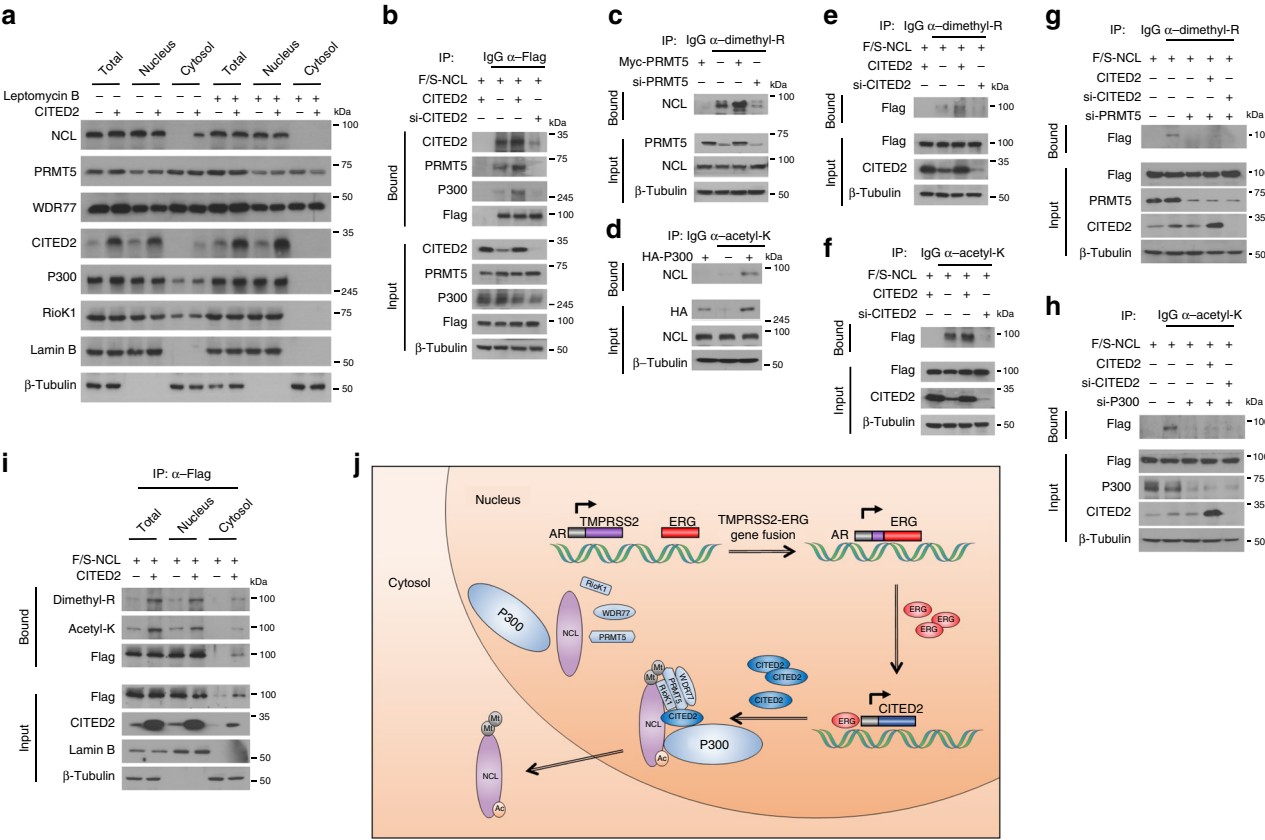

**Fig. 4** CITED2 is essential for post-translational modifications of NCL. **a** HEK293T cells were transfected with pcDNA or CITED2 and treated with Leptomycin B (200 nM), and the cell lysates were fractionated to cytosolic and nuclear components. The cell fractions were immunoblotted with the indicated antibodies. **b** HEK293T cells were cotransfected with Flag/SBP-NCL and CITED2 or si-CITED2. Cell lysates were immunoprecipitated with anti-Flag and immunoblotted with the indicated antibodies. **c** HEK293T cells were transfected with Myc-PRMT5 and/or si-PRMT5. Cell lysates were immunoprecipitated with anti-dimethyl arginine antibody and precipitated NCL was immunoblotted. **d** HEK293T cells were transfected with HA-p300. Cell lysates were immunoprecipitated with anti-acetyl lysine antibody and precipitated NCL was immunoblotted. **e**, **f** HEK293T cells were cotransfected with Flag/SBP-NCL and CITED2 or si-CITED2. Cell lysates were immunoprecipitated with anti-dimethyl arginine or anti-acetyl lysine antibody and precipitated NCL was immunoblotted. **g**, **h** HEK293T cells were cotransfected with the indicated plasmids and siRNAs. Cell lysates were immunoprecipitated with anti-dimethyl arginine or anti-acetyl lysine antibody and precipitated NCL was immunoblotted. **i** HEK293T cells were cotransfected with Flag/SBP-NCL and CITED2, and the cell lysates were fractionated to cytosolic and nuclear components. The cell fractions were immunoprecipitated with anti-Flag antibody and immunoblotted with the indicated antibodies. **j** The ERG–CITED2–PRMT5/p300–NCL pathway in prostate cancer. ERG is upregulated due to the *TMPRSS–ERG* gene fusion and transactivates the *CITED2* gene in prostate cancer cells. Overexpressed CITED2 induces the methylation and acetylation of NCL in the nucleus by recruiting PRMT5 and p300, then modified NCL is translocated to the cytoplasm. All experiments were carried out at three distinct samples

**CITED2 modulates translocation of NCL through methylation and acetylation**. To examine whether CITED2 affects the subcellular localization of these subunits by forming a complex, we evaluated each subunit in the nuclear and cytoplasmic fractions of HEK293T cells overexpressing CITED2. CITED2 overexpression reduced nuclear expression of NCL but enhanced cytoplasmic expression, which was attenuated by a nuclear export inhibitor Leptomycin B (Fig. 4a). This suggests that CITED2 induces the nuclear export of NCL. To determine the role of CITED2 in the PRMT5/p300/NCL complex, immunoprecipitation was performed using HEK293T cells with either CITED2 overexpression or silencing. Notably, PRMT5 and p300 binding to NCL was potentiated by CITED2 overexpression but weakened by CITED2 knockdown (Fig. 4b). This result prompted us to determine whether CITED2 acts as a molecular chaperone to guide PRMT5 and p300 to NCL. As expected, NCL was arginine dimethylated and lysine acetylated by PRMT5 and p300, respectively (Fig. 4c, d). More importantly, both modifications were dependent on

CITED2 (Fig. 4e, f). The CITED2-dependent modifications of NCL were attenuated by PRMT5 and p300 knockdown (Fig. 4g, h), which supports our hypothesis that CITED2 promotes post-translational modifications of NCL by recruiting PRMT5 and p300. Our next objective was to determine the subcellular location where CITED2-facilitated NCL modification occurs. The CITED2-dependent modifications of NCL were detected in the nuclear fraction, which was expected since NCL is present mainly in the nucleus (Fig. 4i). Although the level of NCL protein was low in the cytoplasmic fraction compared to nuclear fraction, surprisingly, the dimethylated and acetylated NCL forms were clearly detected in the same fraction (Fig. 4i). The nuclear export of NCL was promoted by CITED2 overexpression, which was reversed by a PRMT5 inhibitor EPZ015666 (Supplementary Figure 4). These results suggest that NCL is modified in the nucleus and then translocated to the cytoplasm in part. The EGR–CITED2–PRMT5/p300–NCL pathway is summarized in Fig. 4j.

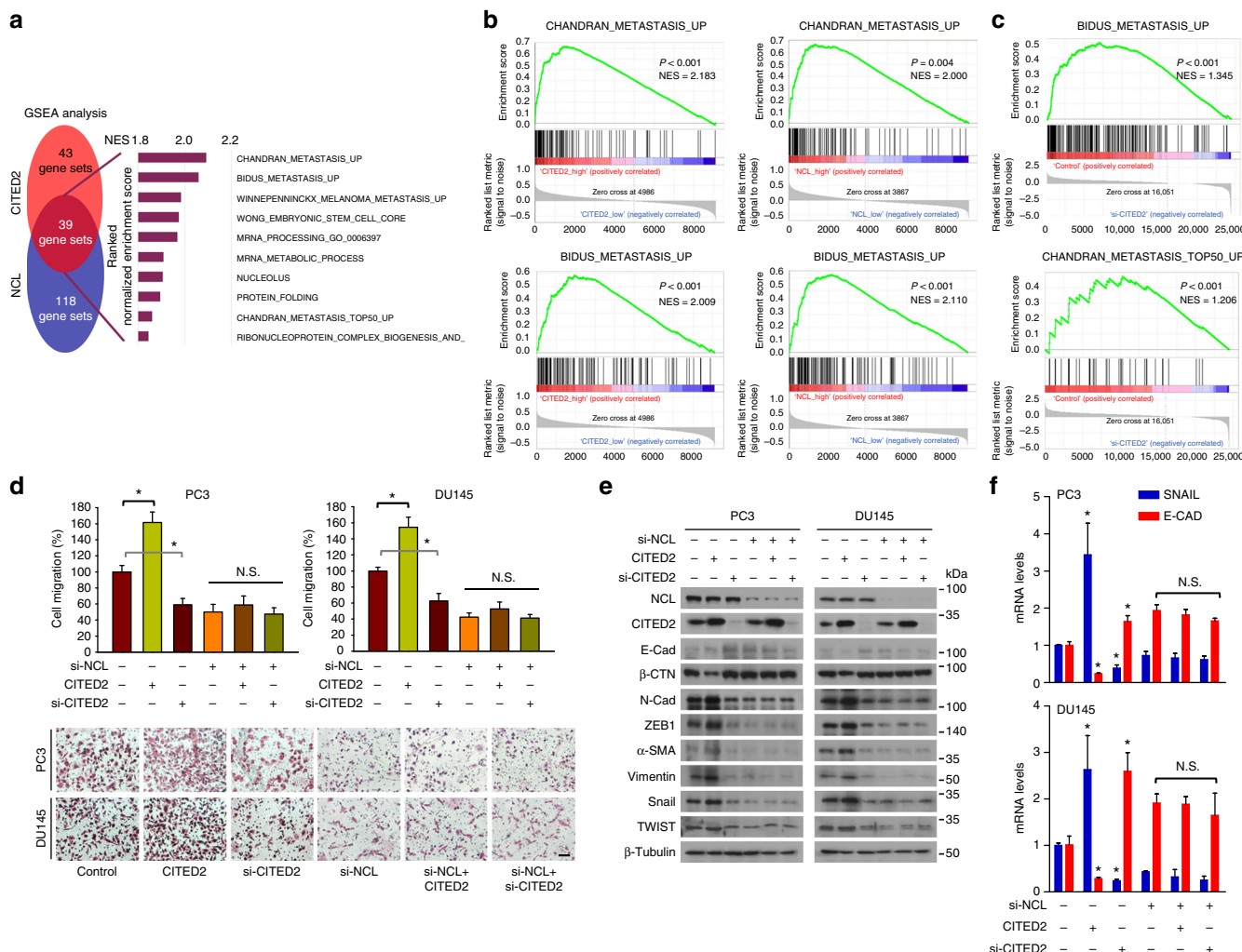

**Fig. 5** CITED2 enhances metastatic potential NCL-dependently in prostate cancer. **a** Gene Set Enrichment Analysis (GSEA). Venn diagram depicts 39 gene sets upregulated ($P < 0.05$ and FDR < 0.30) commonly by CITED2 and NCL expression. Of them, top 10 gene sets are listed. **b** Representative enrichment plots of the metastasis-related gene set which positively correlate with CITED2 (left panel) or NCL (right panel). **c** PC3 cells were transfected with control or CITED2-targeting siRNA. Total RNAs were extracted using Trizol and subjected to RNA-sequencing analyses. The experiments were performed three times independently. The graphs show representative enrichment plots of the metastasis-related gene set which positively correlate with CITED2 in PC3 cell. **d** Cell migration was analyzed using a transwell chamber. PC3 and DU145 cells ($1 \times 10^4$/well), which had been transfected as indicated, were placed on the upper chamber. After 12 h, cells passing through the interface membrane were stained (bottom) and counted (top). Each bar represents the mean + SD ($n = 3$). The scale bar represents 25 μm. **e** PC3 and DU145 cells were transfected with CITED2 or si-CITED2, and/or si-NCL. Representative EMT markers were immunoblotted. **f** RNAs were extracted from PC3 or DU145 cells which were transfected with CITED2 or si-CITED2 and/or si-NCL. The *SNAIL* and *E-CAD* mRNA levels were measured by RT-qPCR. Each bar represents the mean + SD ($n = 3$). *$P < 0.05$ versus the control group; N.S. not significantly different' among the groups by Student's *t*-test

**The CITED2–NCL axis positively regulates epithelial–mesenchymal transition (EMT) and cell migration in prostate cancer.** The cellular consequences of the CITED2–NCL axis were examined using gene set enrichment analyses. Prostate cancer tissues in the GSE6919 data set were divided into low and high expression groups based on the mean CITED2 and NCL expression values (Supplementary Figure 5a). We identified the gene sets that were enriched in the high group compared with the low group (Supplementary Figure 5b). Several metastasis-related gene sets were among the top 10 gene sets enriched in the CITED2_high and NCL_high groups. To determine the role of the CITED2–NCL axis in cellular processes, we searched for gene sets commonly associated with CITED2 and NCL. Five of the top 10 common gene sets were related to metastasis (Fig. 5a). The enrichment profiles of two representative gene sets associated with CITED2

and NCL expression are shown in Fig. 5b and those of other gene sets in Supplementary Figure 5c, d. To support the patient-derived gene set enrichment data, we conducted RNA-sequencing (RNA-seq) analysis in PC3 cells. CITED2 was knocked down in PC3 cells using siRNAs (Supplementary Figure 6a). To observe changes in gene expression pattern by CITED2 knockdown, heat map clustering was performed (Supplementary Figure 6b). We found that metastasis-related gene sets were more enriched in the control group than in the CITED2 knockdown group (Fig. 5c). Based on these results, we evaluated whether the CITED2–NCL axis is involved in prostate cancer cell migration. In phalloidin-stained PC3 and DU145 cells, lamellipodia were formed depending on CITED2 expression (Supplementary Figure 7a). In Transwell® migration and invasion assays, CITED2 stimulated cell migration and invasion in an NCL-dependent manner

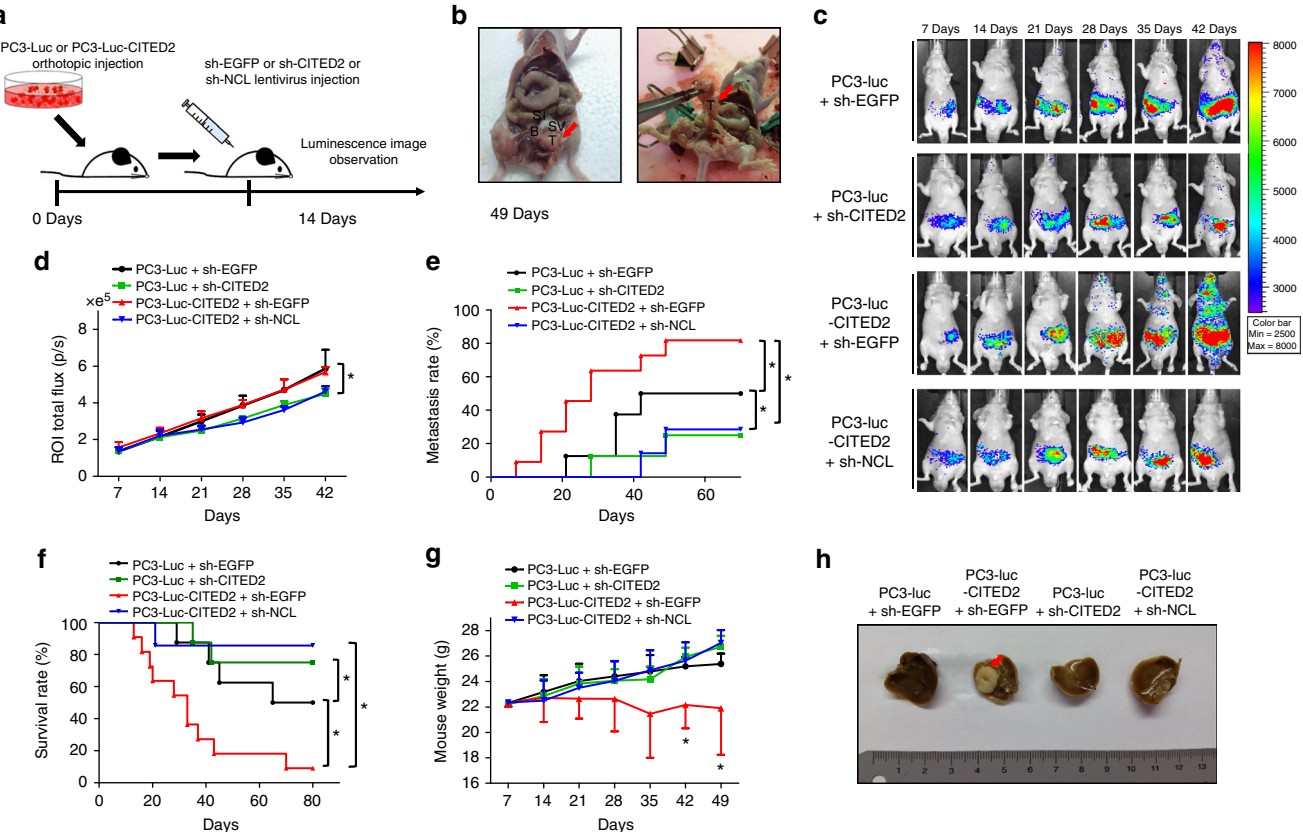

**Fig. 6** CITED2 promotes prostate cancer metastasis NCL-dependently in mice. **a** Schematic diagram of in vivo prostate metastasis model. **b** Representative photographs of prostate tumors 10 weeks after cell implantation. The luciferase-expressing PC3 cells ($0.5 \times 10^6$) were suspended in 20 μL of sterile PBS and injected into the prostates of Balb/cSlc-nu/nu mice. **c** Bioluminescent images of primary tumors and metastases were monitored using Xenogen IVIS® Lumina 100. Color scale bars represent tumor intensity from purple (low) to red (high). **d** Growth curves of primary tumors were plotted based on bioluminescence intensities. Data are presented as the mean + SD, and *$P < 0.05$ between two groups by Mann–Whitney statistical analysis. Mouse numbers are 8 in the sh-EGFP group, 8 in the sh-CITED2 group, 11 in the CITED2+sh-EGFP, and 7 in the CITED2+sh-NCL group. **e** Metastasis rate was retrieved according to the Kaplan–Meier method and *$P < 0.05$ between two groups. Metastasis defined as ROI flux value was larger than $1.0 \times e^5$. **f** Kaplan–Meier overall survival rate analyses were followed up until 80 days after xenograft and *$P < 0.05$ between two groups. **g** Tumor-bearing mice were weighed in the indicated times. Data are presented as the mean + SD, and *$P < 0.05$ between two groups by Mann–Whitney statistical analysis. **h** Representative photographs of livers with metastatic carcinoma nodules (indicated by arrow)

(Fig. 5d and Supplementary Figure 7b). The CITED2 and NCL expression levels in the cells were verified by western blotting (Supplementary Figure 7c). In addition, CITED2 increased the protein and mRNA levels of mesenchymal markers but decreased those of epithelial markers, and these effects of CITED2 were attenuated by knocking down NCL (Fig. 5e, f). However, CITED2 or NCL expression did not affect cell growth or viability in prostate cancer cells (Supplementary Fig 7d–f). Taken together, these results strongly suggest that the CITED2–NCL axis enhances the metastatic potential, rather than cell growth, in prostate cancer cells.

**The CITED2–NCL axis promotes prostate cancer metastasis in mice.** To characterize the in vivo role of the CITED2–NCL axis in metastasis, we established stable PC3 cell lines and implanted them into the prostates of male athymic nude mice (Fig. 6a). CITED2 overexpression (Supplementary Figure 8a), luciferase activity (Supplementary Figure 8b), and the degree of cell migration (Supplementary Figure 8c) were assessed in PC3 stable cell lines. The gene-silencing efficacies of lentiviruses harboring five different small hairpin RNAs (shRNAs) targeting CITED2 or NCL were evaluated by western blotting (Supplementary

Figure 8d). The abdomens of the mice were opened 2 months after cell implantation, revealing strong growth of the xenografted tumors in the prostates (Fig. 6b). In tumor tissue homogenates obtained from mouse xenograft, protein expressions of CITED2 and NCL were measured to verify the overexpression or knock-down of CITED2 and NCL (Supplementary Figure 8e). We monitored the bioluminescence emitted from cancer cells each week to trace the metastatic growth of the prostate tumors (Supplementary Figure 9). Compared with the control group, metastasis was enhanced in the CITED2-overexpressing group but reduced in the CITED2 knockdown group. The metastasis-promoting effect of CITED2 overexpression was abolished by NCL knockdown (Fig. 6c). Integrated values of region of interest (ROI) luminescence were used in statistical analyses of tumor growth and metastasis. The results showed that prostate tumor growth was delayed by CITED2 or NCL knockdown (Fig. 6d). Metastasis was significantly enhanced by CITED2 overexpression, and this effect was reversed by NCL knockdown (Fig. 6e). Furthermore, mouse survival was decreased by CITED2 over-expression but rescued by CITED2 or NCL knockdown (Fig. 6f). The CITED2 overexpression group showed significant body weight loss, suggesting that these mice might be cachectic (Fig. 6g). Representative images of liver metastases are shown in

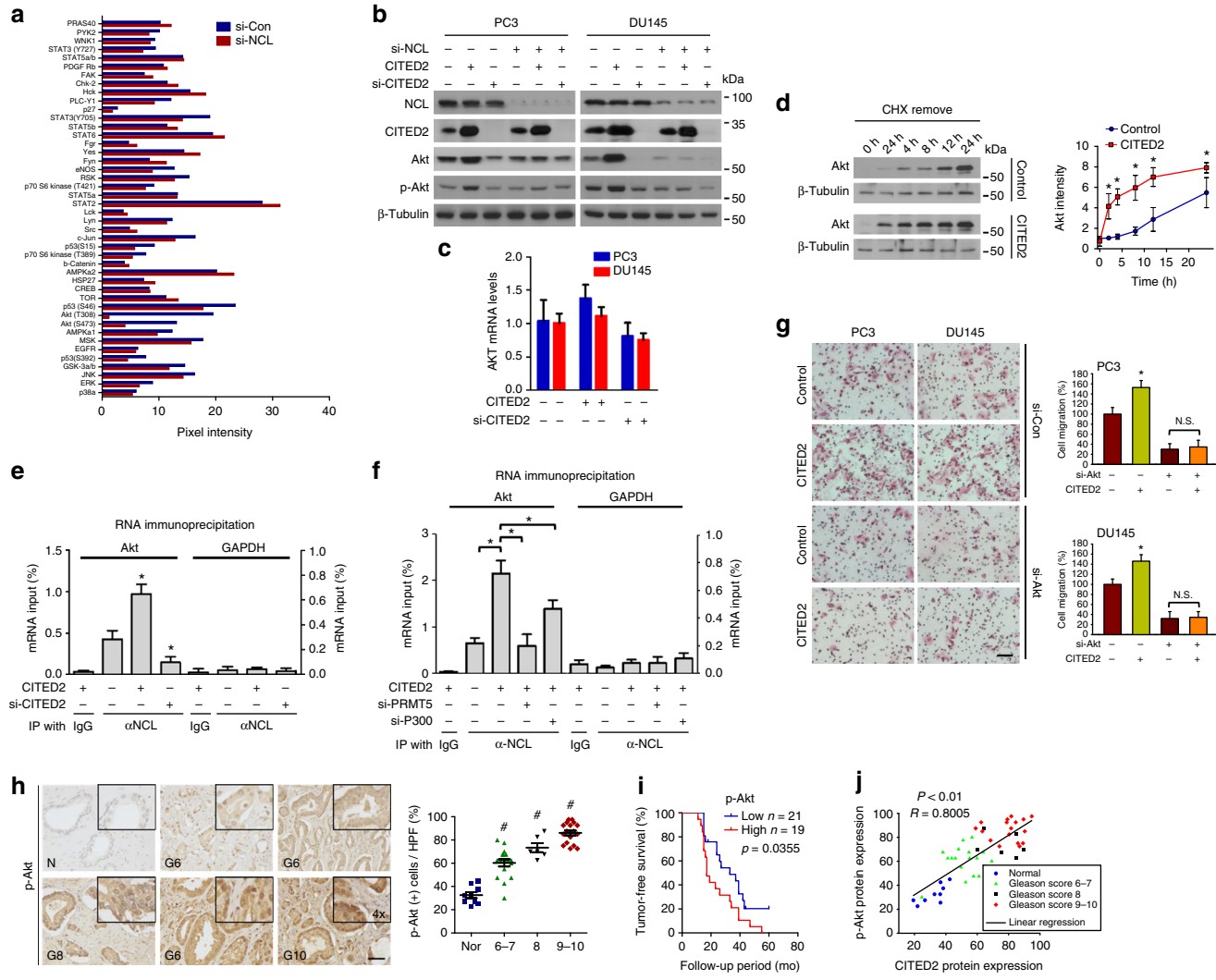

**Fig. 7** CITED2 facilitates AKT synthesis through post-translational modifications of NCL. **a** The blot spot pixel densities were analyzed by the ImageJ program. **b** Immunoblotting of NCL, CITED2, AKT, p-AKT, and β-tubulin in cells that were transfected with CITED2 or si-CITED2 and si-NCL. **c** The AKT mRNA levels in cells were measured by RT-qPCR. **d** DU145 cells, which had been transfected with pcDNA or CITED2, were pretreated with 100 μM cycloheximide for 12 h. After washing out cycloheximide, cells were incubated for the indicated times. AKT synthesis levels were detected by western blotting and quantified using ImageJ. Each point represents the mean ± SD (n = 3). **e**, **f** RNA-IP was performed to analyze the NCL binding to AKT mRNA. DU145 cells were transfected as indicated and the cell lysates were immunoprecipitated with anti-NCL or IgG. The co-precipitated *AKT* or *GAPDH* (a negative control) mRNA was quantified using RT-qPCR. Results (the mean ± SD, n = 3) were represented as percentages of IP signal/input signal (% input). **g** Cells that had been transfected with CITED2 and/or si-AKT were subjected to cell migration assay. Cells passing through the interface membrane were stained and counted. The relative cell migration is presented as a bar graph (the mean + SD, n = 3). N.S. 'not significantly different' among the groups. The scale bar represents 25 μm. **h** Representative images of human prostate adenocarcinoma tissues immunostained with anti-phospho-AKT antibody. Abbreviations at the left bottoms of images: N normal prostate tissue (blue squares), G6–10 (6–7, green triangles; 8, black inverted triangles; 9–10, red diamond) Gleason scores 6–10 of prostate cancer (left panel). The immunostaining scores were calculated by counting stained cells and presented as dot plots (right panel). The horizontal lines in dot plot represent the means ± SE and #P < 0.05 versus the normal group by Mann–Whitney statistical analysis. The scale bar represents 50 μm. **i** Kaplan–Meier tumor-free survival analysis of prostate cancer patients. P value was calculated by log-rank test. **j** Scatter diagrams for p-AKT expression versus CITED2 expression. R value is Pearson's correlation coefficient. *P < 0.05 versus control group by Student's t-test

Fig. 6h. Based on these results, the CITED2–NCL axis may activate signaling pathway(s) that strongly induce cancer metastasis.

**CITED2-activated NCL promotes the AKT-driven EMT by enhancing translation of AKT mRNA.** To explore the signaling pathway responsible for NCL-mediated EMT, we performed proteomic analyses of phosphoproteins and found that phospho-AKT was reduced most in NCL knockdown cells (Fig. 7a and Supplementary Figure 10a). The antibodies used in the

microarray are listed in Supplementary Figure 10b. NCL knockdown downregulated the protein levels of total AKT as well as phospho-AKT in prostate cancer cells (Supplementary Figure 10c), which occurred without any change in AKT mRNA levels (Supplementary Figure 10d). The CITED2–NCL axis was shown to affect the levels of AKT and phospho-AKT (Fig. 7b, c). CITED2 overexpression facilitated de novo synthesis of the AKT protein (Fig. 7d) but did not stabilize the protein (Supplementary Figure 10e). Moreover, mRNA processing genes were enriched in cells with altered CITED2 or NCL expression (Supplementary

Figure 10f). These results prompted us to evaluate whether the CITED2–NCL axis regulates the translation of AKT mRNA. We immunoprecipitated NCL using anti-NCL antibody and quantified the amount of co-precipitated AKT mRNA by quantitative reverse transcription-PCR (RT-PCR). The NCL–AKT mRNA interaction was enhanced by CITED2 overexpression but reduced by CITED2 knockdown (Fig. 7e). Glyceraldehyde-3-phosphate dehydrogenase (GAPDH) mRNA was used as a negative control to verify the specificity of RNA immunoprecipitation. Because the interaction between NCL protein and AKT mRNA was attenuated by silencing PRMT5 or P300 even under CITED2 overexpression (Fig. 7f), CITED2 may have enhanced the NCL–AKT mRNA interaction via PRMT5-mediated methylation and P300-mediated acetylation of NCL. To check whether cell migration was stimulated due to enhancement of AKT translation mediated by CITED2, migration assay was conducted with AKT knockdown and CITED2 overexpression (Supplementary Figure 10g). Compared to the control, CITED2 overexpression increased migration in PC3 and DU145 cells, which was attenuated by AKT knockdown (Fig. 7g). Using three different AKT inhibitors (Wortmannin, LY294002, and MK2206), we examined whether the AKT signaling pathway mediates CITED2-induced cell migration. As previously shown in Fig. 5d, CITED2-dependent cell migration was almost completely attenuated by each of these inhibitors (Supplementary Figure 11). In a similar manner, mRNA markers (Supplementary Figure 12a) and EMT protein (Supplementary Figure 12b) were no longer regulated by CITED2. Because AKT signaling promotes SNAIL expression via nuclear factor (NF)-κB activation[35], we examined whether NF-κB mediates the CITED2–AKT–EMT signaling. CITED2-dependent cell migration and invasion were both abolished by an NF-κB inhibitor Bay-11-7082 (Supplementary Figure 12c). IHC was used to determine if AKT is activated in prostate cancer tissues. Phospho-AKT expression increased concomitantly with an increase in the Gleason score (Fig. 7h) and was associated with poor survival of patients with prostate cancer (Fig. 7i). Pearson's correlation analyses revealed a significant positive correlation between phospho-AKT and CITED2 levels (Fig. 7j).

## Discussion

The current treatments for prostate cancer include surgery, irradiation, and androgen deprivation, but none of these treatments are effective for metastatic castration-resistant prostate cancer (CRPC). Docetaxel is currently used to treat CRPC, because it prolongs the median survival by 3 months[36], and abiraterone is an emerging anti-CRPC drug capable of increasing the survival by 4 months[37]. Unfortunately, the anticancer effects of these drugs are very limited because of the development of drug resistance[38,39]. In this study, CITED2 was found to be uniquely overexpressed in prostate cancer cells, in which it promoted metastasis by activating the NCL–AKT signaling pathway. We therefore propose that CITED2 may be a potential target for treating metastatic prostate cancer.

ERG is an important factor that contributes to prostate cancer progression[4,5]. It is not expressed in normal prostate epithelium but is markedly amplified in prostate cancer because of its gene fusion to the androgen-driven promoter TMPRSS2[40]. Other ETS gene family members, such as *ETV1, ETV4, ETV5,* and *FLI1*, can also fuse to TMPRSS2, but these fusion events display much lower frequencies compared with ERG[41–43]. The significance of ERG gene fusion was demonstrated by its correlation with the clinical phase of patients. ERG expression was positively correlated with the Gleason score in prostate cancer tissues, and it was associated with prostate cancer metastasis and poor patient prognoses[44]. Many

follow-up studies have been conducted to understand how ERG promotes prostate cancer. For example, one study reported that ERG reorganizes actin filaments by activating vimentin and upregulating matrix metalloproteinases, leading to cell invasion[45]. ERG was also reported to facilitate cell movement by inducing the EMT[46]. However, the downstream signaling pathway responsible for ERG-driven metastasis remains unclear. We identified ERG as a transcription factor regulating expression of the *CITED2* gene, which is specifically overexpressed in prostate cancer, and further clarified the ERG–CITED2 axis as the downstream pathway involved in prostate cancer.

Because NCL exists in a complex with PRMT5 and p300, we tested the possibility that NCL is post-translationally co-modified by PRMT5 and p300. Although arginine methylation does not significantly affect the overall charge of NCL, steric hindrance or hydrogen bonds around arginine could be altered. Thus, methylation can modify intermolecular interactions[47–49]. In proteins containing the RNA-binding RGG domain, alterations of protein–RNA interactions by methylation have been reported previously[50]. Because NCL is also an RNA-binding protein with a RGG motif[51], we tested the possibility that NCL binding to AKT mRNA is regulated by the PRMT5-mediated arginine dimethylation of NCL. Methylation enhanced the interaction between NCL and AKT mRNA, thereby facilitating de novo synthesis of AKT translationally. However, lysine acetylation of NCL has not been investigated comprehensively. A previous study reported that p300-mediated lysine acetylation stabilized NCL[52]. However, we observed no change in the level of NCL after overexpressing or knocking down p300. According to another study, acetylation may influence the binding between NCL and nucleic acids. Lysine acetylation is essential for STAT3 (signal transducer and activator of transcription 3) or p53 binding to DNA[53,54]. In a similar manner, we speculate that NCL acetylation by P300 can affect mRNA polysome formation. This possibility needs to be confirmed by additional studies.

Notably, we found that the NCL complex was present mainly in the nucleus, but NCL was bound to AKT mRNA in the cytoplasm. The difference in the location of NCL suggests that NCL is post-translationally modified in the nucleus and then transported to the cytoplasm. Because the entire NCL complex was too large to pass through the nuclear membrane, it is reasonable to assume that the modified NCL is released from the complex and then translocated to the cytoplasm. Consistent with these possibilities, a study demonstrated translocation of NCL from the nucleus to the cytoplasm and the plasma membrane after phosphorylation[55]. However, no study has reported the translocation of NCL after acetylation or methylation, which suggests a mechanism underlying NCL translocation.

Although CITED2 has no special functional domain, it can participate in important biological processes as a scaffolding protein. CITED2 is comprised of three CR(1–3) and one SRJ domains[56,57]. Because each domain provides a docking site for protein interactions, CITED2 with multiple domains may act as a central scaffold recruiting different proteins. For example, CR2 interacts with transcription factors such as TFAP2, HNF4a, PPARa/r, and Smad 2/3 and enhances gene expression by recruiting CBP/p300[6,10,58,59]. CR1 interacts with the GCN5 acetyltransferase, thereby inhibiting the GCN5-mediated acetylation of PGC-1α[60]. CR3 binds to the homeobox protein LHx2 and increases expression of glycoprotein hormone α-subunit[61]. Because we are intrigued by the numerous functions of CITED2, which depend on its binding molecule, we investigated its potential role in tumorigenesis. Previous studies have reported that CITED2 increases cancer progression. It has been reported

that CITED2 promotes MYC-mediated transactivation of the *E2F3* gene by recruiting p300 to stimulate lung cancer progression[16]. However, little is known about the role of CITED2 in cancer metastasis. We thus conducted a screening in patients to identify the cancer type most affected by CITED2 expression and found that CITED2 was most elevated in metastatic prostate cancer. However, considering that a kind of cytokine storm occurs in tumor microenvironment, CITED2 could be differentially expressed in primary and metastatic tumors because they grow with distinct stromal cells. Indeed, various growth factors and cytokines have been reported to increase CITED2 expression[8]. Therefore, we cannot rule out the possibility that CITED2 is overexpressed in metastatic tumor milieu. Nonetheless, our cellular and animal experiments support our notion that ERG-induced CITED2 promotes prostate cancer metastasis. According to this scenario, CITED2 could be a potential target to prevent prostate cancer metastasis. Since the complete inhibition of CITED2 has been reported to induce acute bone marrow failure[14], the anti-CITED2 strategy should be carefully optimized before clinical application.

Because AKT is involved in important oncogenic pathways, most studies have emphasized its role in survival and the cell cycle. AKT activates the mTOR (mammalian target of rapamycin) pathway, which increases cyclin D1 translation to promote cell cycle progression[62], and stimulates CREB activity to induce survival genes such as *Bcl-2*[63]. Many recent studies have also characterized the roles of AKT in EMT and cell migration. AKT not only increases SNAIL expression by activating NF-κB[35], but also stabilizes the SNAIL protein by inactivating glycogen synthase kinase-3β (GSK-3β)[64]. Moreover, AKT enhances transcription of the *SNAIL* and *SLUG* genes by phosphorylating β-catenin[65]. According to past studies, the AKT pathway is aberrantly activated in cancers because of *AKT* gene amplification and the *PTEN* gene deletion[66]. Our study suggests a mechanism involving AKT activation in prostate cancer. The CITED2 stimulation of AKT translation strengthens AKT signaling to promote EMT and eventually cancer metastasis.

In this study, we identified a pathway related to metastasis, involving ERG, CITED2, NCL, and AKT pathway, in prostate cancer. This metastasis-promoting mechanism may be particularly important in prostate cancer overexpressing ERG due to gene fusion events involving ERG. We also identified CITED2 and NCL as target molecules for preventing prostate cancer metastasis in an orthotopic xenograft animal model. Overall, this study provides a basis for future concept studies to develop the next generation of prostate cancer treatments.

## Methods

**Reagents and antibodies**. Antibodies against CITED2, PRMT5, β-tubulin, WDR77, RioK1, p300, β-CTN, Vimentin, TWIST, Snail, N-Cad, and ZEB1 were purchased from Santa Cruz Biotechnology (Santa Cruz, CA); anti-NCL, anti-dimethyl-arginine and anti-acetyl-lysine from Upstate Biotechnology (Lake Placid, NY); anti-p-AKT, anti-AKT and anti-E-cadherin from Cell Signaling (Danvers, MA); anti-FLAG, anti-MYC, and anti-HA from Sigma-Aldrich (St. Louis, MO); and anti-ERG and anti-α-SMA from Abcam (Cambridge, MA). MK2206 was purchased from Selleckchem. Human recombinant proteins of CITED2, PRMT5, P300, and NCL were purchased from Origene. Fetal bovine serum (FBS), dithiothreitol, G418 disulfate salts (G418), EPZ015666, Leptomycin B, Bay-11-7082, LY294002, Cycloheximide, Wortmannin, and others were obtained from Sigma-Aldrich. Sources and dilution factors of antibodies used are summarized in Supplementary Table 1.

**siRNAs and plasmids**. The nucleotide sequences (5' to 3') of siRNAs are; UUAUGUCCUUGGGUGAUAGAUT for CITED2 (NM_006079), AGACUAUAG AGGUGGAAAGAAAGC for NCL (NM_005381), AUGAUGUUGAUAAAGC CU, CGUCCUCAGUUAGAUCCU, and CCACGGUUAAUGCAUGCU for ERG #1–3 (NM_001136154), GGACUGGAAUACGCUAAU for PRMT5 (NM_001039619), GACAAAACCGUGGAAGUA for p300 (NM_001429), CCU CACAGCCCUGAAGUACUCUUC for AKT (NM_005163), and AUGAACGU

GAAUUGCUCAA for non-targeting control. The complementary DNAs (cDNAs) of CITED2, NCL, luciferase, and luciferase-CITED2 were cloned by reverse transcription and PCR using Pfu DNA polymerase, and the cDNAs were inserted into pcDNA, Myc-tagged, FLAG-tagged, HA-tagged, or FLAG/streptavidin-binding protein (SBP)-tagged vectors by blunt-end ligation. TRC lentiviral shRNA targeting CITED2 or NCL were purchased from Dharmacon (Lafayette, CO).

**Cell lines and cell culture**. HEK293T (human embryonic kidney) and human prostate cancer (PC3, DU145, VCaP, LNCaP, C42B, and 22RV1) cell lines were obtained from the American Type Culture Collection (Manassas, VA). Mycoplasma contamination was routinely tested when cell growth or shape was changed. The cell lines were cultured in RPMI-1640 or Dulbecco's modified Eagle's medium supplemented with 10% heat-inactivated FBS in a 5% $CO_2$ humidified atmosphere at 37 °C. Luciferase-expressing and luciferase/CITED2-co-expressing PC3 stable cell lines were established from five G418-resistant clones per cell line. The expression of luciferase has been confirmed with luciferase assay.

**Immunoblotting and immunoprecipitation**. Cell lysates were separated on SDS-polyacrylamide gels, and transferred to Immobilon-P membranes (Millipore, Bedford, MA). Membranes were blocked with a Tris/saline solution containing 5% skim milk and 0.1% Tween-20 for 1 h, and incubated with a primary antibody overnight at 4 °C. Membranes were incubated with a horseradish peroxidase–conjugated secondary antibody for 1 h, and visualized using the ECL kit (Thermo; Rockford, IL). To analyze protein interactions, cell lysates were incubated with anti-CITED2, anti-Flag, or anti-Ac-K or dimethyl-R antibody for 4 h at 4 °C, and the immune complexes were precipitated with protein A/G beads (Santa Cruz, CA). Precipitated proteins were eluted in a denaturing 2× SDS sample buffer, loaded on sodium dodecyl sulfate–polyacrylamide gel electrophoresis (SDS-PAGE), and immunoblotted.

**IHC of human prostate cancer tissue array**. Human prostate cancer tissue arrays were purchased from SuperBioChips Lab (Seoul, South Korea). Clinical information on prostate cancer patients is summarized in Supplementary Table 2. Tumor staging was defined according to the AJCC (American Joint Committee on Cancer) cancer staging manual (7th edition)[67]. The array slides were dried for 1 h in an oven at 60 °C, dewaxed, and autoclaved in an antigen retrieval solution. Tissue sections were treated with 3% $H_2O_2$, and then incubated with a primary antibody (against CITED2, ERG, or p-AKT) overnight at 4 °C, and with a biotinylated secondary antibody for 1 h at room temperature. The immune complexes were visualized using the Vectastatin ABC kit (Vector Laboratories, Burlingame, CA), and tissue slides were counterstained with hematoxylin for 10 min. The immune-stained cells were counted at four high-power fields for each tissue.

**Immunofluorescence**. Cells grown on cover slides were fixed with methanol for 30 min, permeabilized with 0.1% Triton X-100 for 10 min, blocked by 3% bovine serum albumin for 2 h, and then incubated with a primary antibody in the dark overnight at 4 °C. The slides were incubated with Alexa Flour® 488 IgG anti-mouse/rabbit (green, 1:200), Alexa Flour® 568 IgG anti-goat (red, 1:200), Alexa Flour® 647 IgG anti-mouse (purple, 1:200), or Alexa Flour® 633 phalloidin (F-actin) solution in the dark for 1 h. Then, nuclei were stained with 4′,6-diamidino-2-phenylindole (DAPI) for 10 min. Fluorescence images were photographed using confocal microscopy.

**Orthotopic xenograft mouse model**. All animal studies were carried out according to the proposed protocol approved by the Seoul National University Institutional Animal Care and Use Committee (No. 150629-4-1). PC3 prostate cancer cells were transfected with the luciferase-IRES-EGFP or the luciferase-IRES-CITED2 plasmid and treated with G418 to select stable cell lines. Male 8-week-old Balb/cSlc-nu/nu mice were used for orthopotic xenografts. We opened the low midline abdomen of mouse with 3–4 mm incision, and smoothly pressed the bladder using sterile cotton swab to find the prostate. The PC3 stable cell lines were injected into the ventral lobe of prostate. After 14 days, shRNA lentiviruses were injected into grafted tumors, and tumor growth and metastasis were monitored using Xenogen IVIS® Lumina.

**Informatics analysis**. Publicly available prostate cancer microarray data set GSE6919 was analyzed to compare CITED2 and NCL mRNA levels between normal and cancer tissues. All tissues ($n = 171$) were grouped as four classes: normal prostate tissues free of any pathological alteration ($n = 18$), normal prostate tissues adjacent to tumors ($n = 63$), primary prostate tumors ($n = 65$), and metastatic prostate tumors ($n = 25$). The values of the 33113_at probe (corresponding to CITED2), the 32590_at (corresponding to NCL) on each group were calculated and compared between the four groups using Pearson's correlation. The prostate cancer gene set enrichment analysis (GSEA) was also performed using GSE6919 data set, and a formatted GCT file was used as input for the GSEA algorithm v2.0 (available from: http://www.broadinstitute.org/gsea). For grouping the GSE6919 data set, the values of the 33113 or 32590_at probe were used as criteria standard for low expression and high expression group. CITED2 mRNA expression in 28

different types of cancer were obtained from TCGA cancer provisional data sets based on TCGA Research Network (http://cancergenome.nih.gov). The abbreviations used in Fig. 1a and the number of patients are as follows: ACC, adrenocortical carcinoma ($n = 79$); Bladder, bladder urothelial carcinoma ($n = 408$); Glioma, brain lower grade glioma ($n = 530$); Breast, breast invasive carcinoma ($n = 1100$); Cervical, cervical squamous cell carcinoma and endocervical adenocarcinoma ($n = 306$); Cholangiocarcinoma ($n = 36$); Colorectal, colorectal adenocarcinoma ($n = 382$); GBM, glioblastoma multiforme ($n = 166$); Head & neck, head and neck squamous cell carcinoma ($n = 522$); chRCC, kidney chromophobe ($n = 66$); ccRCC, kidney renal clear cell carcinoma ($n = 534$); Liver, liver hepatocellular carcinoma ($n = 373$); Lung adeno, lung adenocarcinoma ($n = 517$); Lung squ, lung squamous cell carcinoma ($n = 501$); DLBC, lymphoid neoplasm diffuse large B-cell lymphoma ($n = 48$); Mesothelioma ($n = 87$); Ovarian, ovarian serous cystadeno-carcinoma ($n = 307$); Pancreas, pancreatic adenocarcinoma ($n = 179$); PCPG, pheochromocytoma and paraganglioma ($n = 184$); Prostate, prostate adenocarci-noma ($n = 498$); Sarcoma ($n = 263$); Melanoma, skin cutaneous melanoma ($n = 472$); Testicular Germ Cell, testicular germ cell cancer ($n = 156$); Thymoma ($n = 120$); Thyroid, thyroid carcinoma ($n = 509$); Uterine CS, uterine carcinosarcoma ($n = 57$); Uterine, uterine corpus endometrial carcinoma ($n = 177$); Uveal mela-noma ($n = 80$).

**Fractionation of cytoplasmic and nuclear components**. Cells were spun down at $800 \times g$ for 5 min, and gently homogenized in a hypotonic solution containing 20 mM Tris/HCl (pH 7.8), 1.5 mM MgCl₂, 10 mM KCl, 0.2 mM EDTA, 0.5% NP-40, 0.5 mM dithiotheritol, and 0.5 mM phenylmethylsulfonyl fluoride (PMSF). The cell lysates were centrifuged at $3000 \times g$ for 10 min at 4 °C, and the supernatant was collected as the cytosolic fraction. The pellet was resuspended in a hypertonic solution containing 20 mM Tris/HCl (pH 7.8), 400 mM NaCl, 1 mM EDTA, 1.5 mM MgCl₂, 10% glycerol, 0.5 mM dithiotheritol, and 0.5 mM PMSF, and intermittently vortexed on ice for 30 min. After the suspension was centrifuged at $18,000 \times g$ for 20 min at 4 °C, the supernatant was collected as the nuclear fraction.

**Cell viability assay**. Cells were grown in 98-well culture plates, and incubated with 100 µL/well of the MTT labeling reagent (Sigma-Aldrich) for 3 h. Blue formazan crystals were solubilized with acidified isopropanol, and formazan levels were determined at 570 nm.

**Fast protein liquid chromatography**. Fast protein liquid chromatography (FPLC) analysis was performed on Preparative Biomolecular Purification System equipped with AKTA explorer 10 and Superdex 200 10/300 GL column (GE Healthcare, Uppsala, Sweden). After transfection with CITED2 or empty vector, the cells were centrifuged at $800 \times g$ for 5 min, and resuspended with a lysis buffer consisting of 20 mM Tris/HCl (pH 7.5), 150 mM NaCl, 1 mM EDTA, 0.5% NP-40, 0.5 mM PMSF and protease inhibitor. The cell lysates were centrifuged at $4000 \times g$ for 10 min to separate into pellet and supernatant. Transfer supernatant and collect it for FPLC analysis. Then, 100 µL of protein elusion was continuously monitored at 280 nm using a UV detector. To estimate molecular weight of proteins in each fraction, the Sigma-Aldrich FPLC protein markers (29–700 kDa) were run on FPLC in the same condition. All procedures were carried out at 4 °C. Each elute was subjected to immunoblotting with antibodies against NCL, PRMT5, WDR77, CITED2, P300, and RioK1.

**Migration and invasion assays**. PC3 or DU145 cells were cultured in 24-well transwell plates with an 8.0 µm polycarbonate membrane which were pur-chased from Corning Life Science (Acton, MA). The lower chamber was filled with a culture medium containing 10% FBS as a chemo-attractant. For cell migration analysis, PC3 or DU145 cells in an FBS-free medium were seeded into the upper chamber and incubated at 37 °C for 12 h. For cell invasion analysis, the polycarbonate membrane was coated with 0.5 mg/mL of Matrigel. Cells on the upper surface of the interface membrane were removed using a cotton swab. Migrating cells on the lower surface of the membrane were stained with hematoxylin and eosin, and counted under an optical microscope at a 100× magnification.

**Quantitative RT-PCR**. Total RNA was isolated using TRIZOL reagent (Invitrogen; Carlsbad, CA), and cDNA synthesis was carried out in a reaction mixture (Pro-mega, Madison, WI) containing M-MLV Reverse Transcriptase, RNase inhibitor, dNTP, and random primers at 46 °C for 1 h. Quantitative real-time PCR on 96-well optical plates was performed in the qPCR Mastermix (Enzynomics, Daejeon, Korea), and fluorescence emitting from dye-DNA complex was monitored in CFX Connect Real-Time Cycler (BIO-RAD, Hercules, CA). The mRNA values of tar-geted genes were calculated relative to GAPDH expression. All reactions were performed in triplicate. The nucleotide sequences of PCR primers are summarized in Supplementary Table 3.

**RNA inmunoprecipitation**. RNA immunoprecipitation (RIP) was conducted using the Magna RIP™ RNA-binding protein immunoprecipitation kit (EMD Millipore,

Billerica, MA). Cells were spun down and homogenized in a RIP lysis buffer containing a protease inhibitor cocktail and RNase inhibitor. After cell lysates were centrifuged at $18,000 \times g$ for 10 min, the supernatant was incubated with IgG or anti-NCL antibody in RIP inmmunoprecipitation buffer overnight at 4 °C, followed by incubation with protein A/G magnetic beads. The immune complexes were precipitated using a magnetic separator, and incubated in a protein degradation buffer containing 10% SDS and proteinase K at 55 °C for 30 min. The samples were mixed with 400 µL of phenol:chloroform:isoamyl alcohol and centrifuged at $18,000 \times g$ for 10 min to separate the phases. The aqueous phase (350 µL) was mixed with 400 µL of chloroform, and centrifuged at $18,000 \times g$ for 10 min. The aqueous phase (300 µL) was mixed with 50 µL of salt solution I/II, 5 µL of pre-cipitation enhancer and 850 µL of absolute ethanol, and centrifuged at $18,000 \times g$ for 30 min at 4 °C. The pellet was washed with 80% ethanol, and resolved in 20 µL of RNase-free water. The level of AKT mRNA in the sample was quantified by RT-qPCR and represented as percentage of IP/input signal (% input). All reactions were performed in triplicate.

**Chromatin immunoprecipitation (ChIP)**. Cells were fixed with 37% formaldehyde at 37 °C for 10 min, treated with 150 mM glycine. Fixed cells were lysed with 0.5% NP-40, and centrifuged at $800 \times g$ at 4 °C for 10 min to collect crude nuclear fraction. Nucleus pellet was incubated with 1% SDS and sonicated to shear genomic DNAs into 300–500 bp fragments. Soluble chromatin complexes were immuno-precipitated with IgG or anti-ERG antibody overnight at 4 °C. Immune complexes were precipitated with protein A/G beads pre-blocked by salmon sperm DNA at 4 °C for 4 h. The beads were sequentially washed with a low salt buffer, a high salt buffer, LiCl wash buffer, and TE buffer. The immunoprecipitation chromatin complexes were eluted in a ChIP direct elution buffer at 65 °C for 30 min and incubated overnight at 65 °C to cross-link chromatin complex. DNAs were isolated by phenol-chloroform-isoamyl alcohol (25:24:1) and precipitated with ethanol and glycogen. The extracted DNAs were resolved in nuclease-free water and analyzed by real-time PCR (95 °C/55 °C/72 °C, 30 s at each phase).

**Statistical analysis**. All data were analyzed using Microsoft Excel 2013 software or Graph pad Prism 5 software, and results were expressed as means and SD from three or more distinct samples. We used the unpaired, two-sided Student's $t$-test or Mann–Whitney $U$-test to compare protein expression level, mRNA expression level, cell viability, ROI flux, and cell numbers. Statistical significances were con-sidered when $P$ values were less than 0.05. In addition, protein or mRNA expression correlations were analyzed using Spearman's $p$ statistic. Survival rate analyses were performed by drawing curves and calculating log-rank $P$ test using the Kaplan–Meier method.

## Data availability

The authors declare that all data supporting the findings of this study are available within the article and its supplementary information files or from the corresponding author upon reasonable request. Raw data file for LC-MS/MS is included in Supplementary Data 1. Raw data files for RNA-seq have been deposited in the NCBI Gene Expression Omnibus database under the accession code GSE119113. Full Western blots are presented in Supplementary Figure 13.

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

## Acknowledgements

This work was supported by grants from the National Research Foundation of Korea (2017015015 and 2017048432).

## Author contributions

J.-W.P. and S.-H.S. designed the study and wrote the manuscript. S.-H.S., G.Y.L., M.L., and J.K. performed cell-based experiments and analyzed biochemical and proteomics data. S.-H.S. and H.-W.S. performed animal studies and analyzed informatics. Y.-S.C. constructed expression vectors and established cell lines.

## Additional information

**Competing interests:** The authors declare no competing interests.

