## [Peer Review File · Nature Communications]

Reviewers' Comments:

Reviewer #1:

Remarks to the Author:

Prostate cancer is the most frequently diagnosed cancer among males. Despite many efforts to combat this disease, the prognosis of patients with advanced prostate cancer remains poor, mainly because of the lack of information on mechanisms that induce metastasis. In the current manuscript Seung-Hyun Shin et al investigated the role CITED2 in prostate cancer metastasis. They found that CITED2 is highly expressed in metastatic prostate cancer, and its expression is correlated with poor survival in patients. The mechanism by which CITED2 exerts its function in this cancer was investigated as well, and found to involve chaperone activity. Thus, the expression of this protein induces formation of a big protein complex, containing nucleolin, p300, PRMT5 and probably other proteins. CITED2 overexpression, and its interaction with nucleolin, was found to lead to cell migration and EMT in tissue culture cells, as well as to promote prostate cancer metastasis in mice.

Overall, this is a comprehensive, well performed and well written manuscript that shed a new light on the important mechanism of the induction prostate cancer metastasis. I have only a few comments that should be addressed before publication. These are as follows:

1) The interactions are generally well-proven, but it is not clear whether CITED2 is indeed required for the attachment of PRMT5 and p300 to nucleolin, or whether the interaction is direct. This can be answered by using in vitro binding of recombinant proteins.

2) The experiments that show localization in general, and mainly the export to the cytosol, are not always convincing. In particular, the very small amount of cytosolic complex can be due to proteins that have just been translated or possibly leakage from the nuclear fractions. Proximity-ligation assay (PLA), as well as studying the localization in the presence of leptomycin B (nuclear export inhibitor) could be used to confirm these points.

3) The suggestion that Akt is regulated by translation and not by phosphorylation is problematic. First, it is unlikely that the small amount of complex in the cytosol (if any) may cause such a regulation. Second, the reduction of p-Akt upon knocking down CITED2 or nucleolin is more impressive than the reduction in Akt expression (Fig.7). Therefore, it is likely that the phosphorylation of Akt may be regulated separately from that of Akt translation. These points should be further clarified. Some experiments that can be used for such studies are: showing the localization of p-AKT in the examined cells, useage of AKT inhibitor as compared to that of wortmannin, to show whether the activity of Akt is important here, and using PLA to show interaction of p-Akt and Akt to CITED2.

4) Four minor points:

a) In the model (Fig. 4I) p300 should appear bigger.

b) In Fig S4C, the red box is misplaced (should be in the lane to the right), and is not mentioned in the legend.

c) The results in Fig. S11b do not seem to be representative as the reductions seen in the bar graph is much more impressive than those depicted in the pictures.

d) In Fig. S12A, the natures of the squares and the writing within each panel should be specified.

Reviewer #2:

Remarks to the Author:

The manuscript by Shin et al. described at the molecular level how a CBP/p300 interacting transactivator with E/D-rich carboxy-terminal domain 2, CITED2, plays a new role in prostate cancer (PC) metastasis. The authors presented extensively collected data from gene expression,

patient survival, co-immunoprecipitation (IP), and molecular studies and concluded that CITED2-nucleolin axis is responsible for stimulating cell migration, EMT, and metastasis of PC in mice and thus CITED2 could be a new molecular target for preventing cancer metastasis.

While this manuscript is well-written, described clearly in a very dense manner the IP studies using either overexpressed or knockdown cells to support their conclusions, the manuscript did not focus specifically the biology of CITED2, its interaction with nucleolin in a pathophysiologic meaningful way linking convincingly CITED2-nucleolin axis to PC invasion and metastasis. Specific comments are:

1. TCGA data in figure 1 suggests a role of CITED2 in PC but not other cancer types. When examine PC TCGA data, CITED2 expression can be greatly varied among specimens by as much as 800-fold. A more substantial difference in CITED2 expression seems to exist between PC tumor and tumor metastases, but CITED2 expression is similar when compared between PC and normal tissues adjacent to PC tumors, suggesting factors in the tumor microenvironment may be responsible for the induction of the expression of CITED2. The manuscript did not provide a rationale and the pathophysiologic significance where endogenous CITED2 expression may be regulated that ultimately contribute to cancer metastasis.
2. The authors did not establish fully the levels of CITED2 expression, in a pathophysiologic relevant way, that could be associated with prostate cancer progression and metastasis. For example, if the model described in Figure 4i is correct, the authors need to present data using VCaP cell line with intrinsic TMPRSS2-ERG gene translocation to provide the status of ERG, CITED2 expression and the status of nucleolin binding to cofactors, its posttranslational modification, and partition between cytosol and nuclei in this cell line. It seems that using VCaP to study PC metastasis is more relevant than PC3 due to the presence of AR, and TMPRSS2-ERG gene fusion in the former and the lack of AR and TMPRSS2-ERG gene fusion in PC3 cell line.
3. Careful examination of the data support the possibility that there is no link between CITED2 expression and TMPRSS2 gene fusion. Though the manuscript asserts that "CITED2 is overexpressed in prostate cancer because of TMPRSS2-ERG gene fusion" (line 116), the cause-effect relationship between CITED2 expression and TMPRSS2 gene fusion is sketchy. In Figure 2c, western blotting showed no correlation between ERG expression with CITED2 level. Out of 6 cell lines, only 3 express both ERG and CITED2, while the other 3 cell lines did not show any association between the two genes. Additionally, though Figure 2k showed a high frequency of TMPRSS2-ERG gene fusion in clinical prostate cancer specimens, the same specimens were not further examined for ERG and CITED2 expression and the impact on CITED2-nucleolin axis.
4. Conceptually, it is difficult to provide convincing evidence that the pathway defined by the authors could be responsible for PC progression and metastasis. CITED2 is a known molecular chaperone, and binds many proteins in HEK293T cells as shown in Figure 3. Because CITED2 binds many proteins, it is difficult to assign its role in PC to the interaction with a singular or a selected number of proteins.

Recommendation: Major revision.

Reviewer #3:

Remarks to the Author:

The manuscript by Shin et al. focuses on the role of CITED2 in human prostate cancer. The authors use a combination of molecular and cell biology and in vivo approaches to interrogate the CITED2-nucleolin (NCL)-AKT pathway in prostate cancer. In general, the manuscript is well written. There are however several major technical and conceptual issues which need to be addressed.

- 1) The CITED2 literature is summarised superficially and this needs to be expanded. For example, there are several papers implicating CITED2 in embryonic and adult stem cell functions.

Furthermore, CITED2 is necessary for leukemia biology or in general in cell proliferation. These aspects are not described in sufficient detail.

2) The paragraphs in the introduction appear to be disjoint. The first paragraph is about CITED2, the second about NCL and the third one briefly introduces prostate cancer. However, these paragraphs are not linked together.

3) The authors state: "Because CITED2 functions to inhibit the transcriptional activities of HIFs, it is reasonable that a higher CITED2 level renders RCC less aggressive because of lower expression of HIF target genes." Please be specific here - does this comment relate to HIF-1, HIF-2, HIF-3 or all HIFs. To my knowledge there is no strong evidence that CITED2 controls HIF-2 and HIF-3 activity. Given that HIF-2 is a predominant HIF in RCC, the mechanism the authors suggest is unlikely. It is impossible to make such statement without performing any experiments proving or disproving this suggestion.

4) The data shown on Figure 1C suggest that patients with higher CITED2 levels may have worse overall survival. Please comment on the clinical relevance of this finding, this difference is very small.

5) Regarding Figure 2F, it will be important to quantify the data and present them as enrichment as a % of input. It would also be important to show a negative control for ERG IP, i.e. ChIP-qPCR on CITED2 promoter region which is not bound by ERG.

6) The screen for CITED2 interactors has been performed in HEK293T (embryonic kidney cells) - please provide the justification for using this cell type given that the main focus of the paper is the role CITED2 in prostate cancer. It is surprising the majority for further validation experiments are also performed in HEK293T cells.

7) IP experiments in Figure 3D have some technical issues. Anti-CITED2 IP in LNCaP cells results in a very weak CITED2 band, in fact it is hardly detectable. Given that CITED2 is expressed at weak levels even in the input, it is difficult to explain why anti-CITED2 IP pools down high levels of its potential interactors. Furthermore, anti-CITED2 IP in PNT2 cells showed no interaction with PRMT5. Please clarify and provide more convincing WBs.

8) Figure 3E shows immunofluorescence images in HEK293T cells. Please show equivalent images in prostate cancer cell lines.

9) HA-p300 WB in Figure 3F seems to show several bands. Please show a positive control for this blot and indicate a specific band and non-specific bands in an image which is not so extensively cropped.

10) The number of replicate experiments needs to be stated for all IP experiments and molecular weights need to be shown for WBs.

11) Regarding the statements: "PRMT5, and NCL were identified to bind to the transactivation domain (TAD), the serine/glycine-rich junction (SRJ), and the cysteine/arginine-rich domain 3 (CR3) of CITED2, respectively (Fig. 3F, G)" - these are very artificial experiments. It would be essential to generate specific deletion constructs lacking the indicated domains and demonstrate that this abolishes the claimed interactions. At the moment it is difficult to determine whether CITED2 binds all the listed interactors directly or indirectly (i.e. is a part of a large complex with contains P300/CBP, as published previously, and other interactors).

12) The graphic summary shown in Figure 4I suggests that CITED2 is not expressed "without gene fusion" (left panel). However, CITED2 is expressed in HEK293T cells and therefore this graph is unlikely to be correct. Furthermore, most of the IP experiments were performed in HEK293T cells and therefore it is inappropriate at this stage to extend these data to prostate cancer cells. The right panel suggests that CITED2-containing complexes can be found in the cytosol - what is the experimental evidence that CITED2 is present in the cytosol?

13) GSEA shown in Figure 5A cannot be interpreted as indicative of CITED2 functions. Although CITED2 expression may be increased in the interrogated cells, many genes can be overexpressed too. It would have been more informative to knock down CITED2 expression followed by RNA-seq analyses.

14) Data shown in Figure 5C are not convincing. No striking differences can be seen between Control and CITED2 cells. The data are not quantified using a larger number of cells; the images show several cells. Major conclusions cannot be drawn from this experiment.

15) The authors conclude that CITED2 may be a therapeutic target for prostate cancer treatment.

However, CITED2 inhibition, if this can be pharmacologically achieved, is likely to have severe consequences, e.g. acute bone marrow failure. This has not been discussed at all.

Reviewer #4:

Remarks to the Author:

This study claims that: 1) CITED2 is highly expressed in metastatic prostate cancer because of TMPRSS2-ERG gene fusion, which promoted metastasis by activating NCL at the post-translational level, and 2) CITED2-nucleolin axis stimulate cell migration by inducing the epithelial–mesenchymal transition promoting prostate cancer metastasis in mice. This study shows a novel molecular mechanism by which CITED2 regulates the oncogenic activity of NCL. In general, the experimental methods used to test the hypotheses were correctly applied and the results support the most of the main conclusions.

However, this study requires some further evidence to strengthen and validate some conclusions and explain some results before to be accepted for publication

CITED2 is overexpressed in prostate cancer because of TMPRSS2-ERG gene fusion

1) Lines 125-127. The data in the Fig. 2C do not support that CITED2 protein was highly upregulated in prostate cancer cell lines expressing ERG. This data show only a possible correlation between CITED2 and ERG expression. Correct this strong statement “Because the CITED2 protein was highly upregulated in prostate cancer cell lines expressing ERG (Fig. 2C), we further examined the role of ERG in CITED2 expression.” LNCaP, DU145, and 22Rv1 do not harbor TMPRSS2 – ERG fusion. Why were not tested prostate cancer cell line harboring TMPRSS2 – ERG gene fusion?

2) The data indicate a strongly correlation between tumor harboring TMPRSS2-ERG gene fusion and high expression of CITED2. One of the main claims in this study is that TMPRSS2-ERG gene fusion is a driver to increase the expression of CITED2 in metastatic prostate cancer. These data presented for the authors do not support that TMPRSS2-ERG fusion protein promotes overexpression of CITED2. The authors might use cell lines that harbors TMPRSS2–ERG fusion (such as VCap) and knockdown the expression of wild-type ERG-type to determine the effect on CITED2 expression in presence or absence of androgen.

CITED2 binds to a multimeric complex consisting of NCL, p300, and PRMT5.

1) The Fig 3d shows endogenous PRMT5, NCL, WDR77, RIOK1, and CITED2 interact to form a multicomplex in PC3, DU145, LNCap and PNT2 prostate cell lines by immunoprecipitation of CITED2 and analyzing co-immunoprecipitation of the proteins of interest. In the Fig. 2d is showed that these cells do not express (LNCap and PNT2) or express very low protein levels of CITED2 (PC3 and DU145) compared with 22RV1 and C42B cell lines. How using these negative CITED2-LNCap and PNT2 cell lines there are the co-immunoprecipitation of proteins of interest? Justify and explain this discrepancy.

2) Supplementary Fig. 4d shows an important finding that suggests that CITED2 might regulate the nuclear translocation of NCL to the cytoplasm. Fig. 3e shows only that all the components of the protein multicomplex are localized in the nucleus. Fig. 3e should be showed in the supplementary section and Fig. 4d show in paper.

CITED2 is essential for methylation and acetylation of NCL.

1) Fig. 4a shows clearly that CITED2 overexpression increases the formation of the complex PRMT5/ NCL, but not the complex p300/NCL. Correct the statement or showed convincing data.

2) Are NCL acetylation and methylation required for NCL nucleo-cytoplasmic translocation?

3) Does PRMT5 inhibitor block NCL nucleo-cytoplasmic translocation?

The CITED2-NCL axis positively regulates EMT and cell migration in prostate cancer.

1) Fig. 4d. NCL knockdown after 2-4 days results in slow growth and cell death. Indicate the effect of NCL knockdown on cell viability in these cells prior cell migration and invasion assays. Indicate what time after post-transfection of siRNA NCL the cells were tested.

The CITED2-NCL axis promotes prostate cancer metastasis in mice.

1) Show that the tumors infected with NCL and CITED2 shRNAs have a significant decrease in expression of these proteins compared to the tumors infected with sh-EGF.

CITED2-activated NCL promotes the AKT-driven EMT by enhancing translation of AKT mRNA.

1) Line 241-243. No makes sense the statement "Because CITED2 overexpression was attenuated by PRMT5 knockdown (Fig. 7J), CITED2 may have enhanced the NCL-AKT mRNA interaction via PRMT5-mediated" There are not data that show that PRMT5 knockdown decreases the expression of CITED2 or to support that CITED2 may have enhanced the NCL-AKT mRNA interaction via PRMT5-mediated. Please, show data or correct this statement.

2) Does NCL post-translation modification by PRMT5 modulate the interaction NCL-AKT mRNA?

3) Wortmannin and LY29002 are inhibitors of PI3K. AKT is the dominant effector of PI3K signaling and consequently PI3K inhibitors block the AKT activation. PI3K has other important effectors that contribute to cancer progression such as PDK1-mTORC2-SGK and Rac signaling. Therefore, to determine that AKT signaling pathway mediates CITED-induced migration and EMT is required knockdown the expression of AKT.

4) The figure 7k and l are missing important controls: 1) cells over expressing CITED2 without inhibitor treatment 2) cells transfected with siRNA CITED2 without inhibitor treatment. Indicate if the sample labeled without inhibitor, CITED2 and siCITED2, the cells were transfected with the empty expression plasmid and scramble siRNAs.

Others

1) It is required the author provide detail information about the sources and catalog numbers of antibodies used in each specific immunologic assay (WB, IP, immunofluorescence, RNA precipitation, ChIP assays etc.)

2) Discussion is too long

Answers to Reviewers' Comments

Reviewer #1 (Expertise: Akt signalling, cancer, Remarks to the Author):

Prostate cancer is the most frequently diagnosed cancer among males. Despite many efforts to combat this disease, the prognosis of patients with advanced prostate cancer remains poor, mainly because of the lack of information on mechanisms that induce metastasis. In the current manuscript Seung-Hyun Shin et al investigated the role CITED2 in prostate cancer metastasis. They found that CITED2 is highly expressed in metastatic prostate cancer, and its expression is correlated with poor survival in patients. The mechanism by which CITED2 exerts its function in this cancer was investigated as well, and found to involve chaperone activity. Thus, the expression of this protein induces formation of a big protein complex, containing nucleolin, p300, PRMT5 and probably other proteins. CITED2 overexpression, and its interaction with nucleolin, was found to lead to cell migration and EMT in tissue culture cells, as well as to promote prostate cancer metastasis in mice.

Overall, this is a comprehensive, well performed and well written manuscript that shed a new light on the important mechanism of the induction prostate cancer metastasis. I have only a few comments that should be addressed before publication. These are as follows:

1) The interactions are generally well-proven, but it is not clear whether CITED2 is indeed required for the attachment of PRMT5 and p300 to nucleolin, or whether the interaction is direct. This can be answered by using *in vitro* binding of recombinant proteins.

Revision: We performed a set of *in vitro* binding assays using recombinant proteins of NCL, CITED2, P300 and PRMT5 to examine the interactions between each protein (Fig.3d). As a result, CITED2 directly interacts with NCL, P300 and PRMT5, while PRMT5 does not interact with NCL and P300. P300 directly interacts with NCL, but this interaction was weaker than the interaction between P300 and CITED2. Text revision: lines 27 and 28 on p6; line 1 on p7.

2) The experiments that show localization in general, and mainly the export to the cytosol, are not always convincing. In particular, the very small amount of cytosolic complex can be due to proteins that have just been translated or possibly leakage from the nuclear fractions. Proximity-ligation assay (PLA), as well as studying the localization in the presence of leptomycin B (nuclear export inhibitor) could be used to confirm these points.

Revision: Of two options, we would like to perform the latter one because we have no experience about the PLA assay. As suggested, we used a nuclear export inhibitor leptomycin B to further clarify the subcellular localization. In the presence of leptomycin B, the cytoplasmic levels of CITED2, P300, RioK1, and NCL were markedly reduced, suggesting that the proteins detected in the cytosolic fraction are exported from the nucleus, rather than contaminants of nuclear proteins. In addition, the nuclear and cytosolic levels of PRMT5 and WDR77 were not affected by leptomycin B treatment, suggesting that both proteins are naturally present in both sites. The corresponding data can be found in Fig.4a. Text revision: lines 11 to 15 on p7.

3) The suggestion that Akt is regulated by translation and not by phosphorylation is problematic. First, it is unlikely that the small amount of complex in the cytosol (if any) may cause such a regulation. Second, the reduction of p-Akt upon knocking down CITED2 or nucleolin is more impressive than the reduction in Akt expression (Fig.7). Therefore, it is likely that the phosphorylation of Akt may be regulated separately from that of Akt translation. These points should be further clarified. Some experiments that can be used for such studies are: showing the localization of p-AKT in the examined cells, useage of AKT inhibitor as compared to that of wortmannin, to show whether the activity of Akt is important here, and using PLA to show interaction of p-Akt and Akt to CITED2.

Answer 1: CITED2 expression or NCL knockdown affects p-Akt and total Akt levels to the similar degree because the ratio of p-AKT to AKT was not increased along with CITED2 or NCL expression. This means that the increase in p-AKT results from the increase in total AKT. However, we understand that the western blot in Fig. 7e raised such a concern. The p-AKT blot in Fig.7e is replaced with another one with different film exposure time, which shows the effect of si-CITED2/si-NCL on AKT expression more clearly. Accordingly, we did not further concern the NCL/CITED2-dependent phosphorylation of AKT.

Answer 2: As mentioned in the manuscript, we proposed that CITED2 regulates NCL and NCL targets AKT mRNA to stimulate de novo synthesis of AKT protein. Given this scenario, we thought that we do not need to check the CITED2-AKP(p-AKT) interaction. Indeed, when we checked this interaction using IP, p-AKT and AKT were not co-precipitated with CITED2 (the below figure). Thus, we do not concern this possibility any

more.

4) Four minor points:

a) In the model (Fig. 4I) p300 should appear bigger. :

Revision: Fig.4i in the previous manuscript was moved to Fig.4k. The graphical summary in Fig.4k is modified to reflect the size of p300 more appropriately.

B) In Fig S4C, the red box is misplaced (should be in the lane to the right), and is not mentioned in the legend. :

Answer: Fig.S4C was moved to Fig.3c, as suggested by another reviewer. The red box is to indicate that CITED2 overexpression increases the amount of larger-sized complex (600 ~ 700 kDa).

Revision: The red box is now mentioned in the corresponding figure legend. Text revision: lines 10 to 11 on p25.

C) The results in Fig. S11b do not seem to be representative as the reductions seen in the bar graph is much more impressive than those depicted in the pictures. :

Answer and Revision: Fig.S11b was moved to Fig.S12c. We replaced the images with more representative ones.

D) In Fig. S12A, the natures of the squares and the writing within each panel should be specified.

Answer and Revision: Fig.S12a was moved to Fig.7l, as suggested by a reviewer. The images within smaller boxes in Fig.7l are 4 times magnified ones of the original images to show nucleus and cytosol more clearly. Magnification is indicated in the box.

Reviewer #2 (Expertise: Prostate cancer, metastasis, Remarks to the Author):

The manuscript by Shin et al. described at the molecular level how a CBP/p300 interacting transactivator with E/D-rich carboxy-terminal domain 2, CITED2, plays a new role in prostate cancer (PC) metastasis. The authors presented extensively collected data from gene expression, patient survival, co-immunoprecipitation (IP), and molecular studies and concluded that CITED2-nucleolin axis is responsible for stimulating cell migration, EMT, and metastasis of PC in mice and thus CITED2 could be a new molecular target for preventing cancer metastasis.

While this manuscript is well-written, described clearly in a very dense manner the IP studies using either overexpressed or knockdown cells to support their conclusions, the manuscript did not focus specifically the biology of CITED2, its interaction with nucleolin in a pathophysiologic meaningful way linking convincingly CITED2-nucleolin axis to PC invasion and metastasis. Specific comments are:

1. TCGA data in figure 1 suggests a role of CITED2 in PC but not other cancer types. When examine PC TCGA data, CITED2 expression can be greatly varied among specimens by as much as 800-fold (“point A”). A more substantial difference in CITED2 expression seems to exist between PC tumor and tumor metastases, but CITED2 expression is similar when compared between PC and normal tissues adjacent to PC tumors, suggesting factors in the tumor microenvironment may be responsible for the induction of the expression of CITED2 (“point B”). The manuscript did not provide a rationale and the pathophysiologic significance where endogenous CITED2 expression may be regulated that ultimately contribute to cancer metastasis. :

Point A - Revision: In the box and whiskers plot in Fig.1a, the ‘whiskers’ represent the lowest and highest values

of mRNA expression. For this reason, several outlying values in TCGA data for each cancer types are shown in large deviation and whiskers which are too long, which may make readers misunderstand results. Thus, we replaced the graph with whiskers set to 10-90 percentile. In addition, it is possible that the data variation is different between TCGA and GEO data because TCGA is based on RNA-Seq data but GEO on cDNA array data. Point B - Revision: As was suggested, it is not easy to a high expression of CITED in metastatic tumors is a cause or result for metastasis in this informatics data. However, the following experiments showing the CITED2-induced cell migration and in vivo metastasis clearly supported the possibility that a high expression of CITED2 results in metastasis. Nonetheless, there is another possibility that CITED2 expression is induced under metastatic tumor microenvironment and this is newly mentioned in the Discussion section. Text revision: lines 24 to 28 on p12; lines 1 to 3 on p13.

2. The authors did not establish fully the levels of CITED2 expression, in a pathophysiologic relevant way, that could be associated with prostate cancer progression and metastasis. For example, if the model described in Figure 4i is correct, the authors need to present data using VCaP cell line with intrinsic TMPRSS2-ERG gene translocation to provide the status of ERG, CITED2 expression and the status of nucleolin binding to cofactors, its posttranslational modification, and partition between cytosol and nuclei in this cell line. It seems that using VCaP to study PC metastasis is more relevant than PC3 due to the presence of AR, and TMPRSS2-ERG gene fusion in the former and the lack of AR and TMPRSS2-ERG gene fusion in PC3 cell line. :

Revision: We agree to this comment that VCaP is better to validate our study. We purchased VCaP from ATCC and performed the key experiments in Fig.2. Cell line screening in Fig.2c now includes VCaP cell line, which confirms that TMPRSS2-ERG fusion results in elevated expressions of ERG and CITED2. Western blotting of ERG knockdown in Fig.2d, RT-qPCR in Fig.2e, and androgen response of ERG and CITED2 expression in Fig.2f were performed in VCaP. The binding between ERG and CITED2 promoter was also confirmed in ChIP (Fig.2g) and reporter assay (Fig.2h) in VCaP. In addition, Fig.4i (previous Fig.4k) and the graphical summary was revised. The reason why we used PC3 and DU145 in cell migration experiments was to examine the biological function of CITED2. Actually, androgen can modulate the expression of diverse genes. Although the ERG expression from the TMPRSS2-ERG fusion gene must be induced by androgen, we could not mention that the ERG-CITED2 axis is the main signaling responsible for metastasis because many other genes are also robustly expressed under androgen stimulation. Text revision: lines 16 to 18 on p5; lines 20 to 24 on p5.

3. Careful examination of the data support the possibility that there is no link between CITED2 expression and TMPRSS2 gene fusion. Though the manuscript asserts that “CITED2 is overexpressed in prostate cancer because of TMPRSS2-ERG gene fusion” (line 116), the cause-effect relationship between CITED2 expression and TMPRSS2 gene fusion is sketchy. In Figure 2c, western blotting showed no correlation between ERG expression with CITED2 level. Out of 6 cell lines, only 3 express both ERG and CITED2, while the other 3 cell lines did not show any association between the two genes. Additionally, though Figure 2k showed a high frequency of TMPRSS2-ERG gene fusion in clinical prostate cancer specimens, the same specimens were not further examined for ERG and CITED2 expression and the impact on CITED2-nucleolin axis.:

Revision: In the previous result, ERG or CITED2 level in some cell lines was too weak to examine the correlation between ERG and CITED2 expression. Thus, we carefully re-examined the protein levels and exposed the x-ray film for a longer time to detect them more sensitively (Fig. 2c). With regard to CITED2 and ERG expressions in clinical prostate cancer specimen in Fig.1e and Fig.2i, we showed in Fig.2m that CITED2 expression is much higher in specimens with TMPRSS2-ERG gene fusion.

4. Conceptually, it is difficult to provide convincing evidence that the pathway defined by the authors could be responsible for PC progression and metastasis. CITED2 is a known molecular chaperone, and binds many proteins in HEK293T cells as shown in Figure 3. Because CITED2 binds many proteins, it is difficult to assign its role in PC to the interaction with a singular or a selected number of proteins.

Answer: Molecular chaperones, even enzymes and transcription factors, make a network in protein interactions. CITED2 is one of these cases. It is not strange that CITED2 interacts with diverse proteins. Therefore, to specify the role of NCL in CITED2 functions, we examined whether the CITED2 effect on metastasis is eliminated by silencing NCL. Indeed, I do not know better approaches than NCL O/E and K/D experiments.

Reviewer #3 (Expertise: CITED2, Remarks to the Author):

The manuscript by Shin et al. focuses on the role of CITED2 in human prostate cancer. The authors use a combination of molecular and cell biology and in vivo approaches to interrogate the CITED2-nucleolin (NCL)-AKT pathway in prostate cancer. In general, the manuscript is well written. There are however several major technical and conceptual issues which need to be addressed.

1) The CITED2 literature is summarised superficially and this needs to be expanded. For example, there are several papers implicating CITED2 in embryonic and adult stem cell functions. Furthermore, CITED2 is necessary for leukemia biology or in general in cell proliferation. These aspects are not described in sufficient detail.

Revision: We agree to this comment. We added more information on CITED2 with appropriate references in a new version manuscript. Text revision: lines 16 to 18 on p3.

2) The paragraphs in the introduction appear to be disjoint. The first paragraph is about CITED2, the second about NCL and the third one briefly introduces prostate cancer. However, these paragraphs are not linked together.

Revision: We revised the manuscript to make these paragraphs more linked. Text revision: Introduction on p3 and p4.

3) The authors state: "Because CITED2 functions to inhibit the transcriptional activities of HIFs, it is reasonable that a higher CITED2 level renders RCC less aggressive because of lower expression of HIF target genes." Please be specific here - does this comment relate to HIF-1, HIF-2, HIF-3 or all HIFs. To my knowledge there is no strong evidence that CITED2 controls HIF-2 and HIF-3 activity. Given that HIF-2 is a predominant HIF in RCC, the mechanism the authors suggest is unlikely. It is impossible to make such statement without performing any experiments proving or disproving this suggestion.

Revision: We agree to this comment. The corresponding statement was deleted in the revised manuscript.

4) The data shown on Figure 1C suggest that patients with higher CITED2 levels may have worse overall survival. Please comment on the clinical relevance of this finding, this difference is very small.

Revision: As the total death numbers in the dataset were very low, the difference between the two groups might be also small. Nonetheless, a statistical analysis shows a definite difference. Since we could not find out other TCGA data bases on survival rate in prostate cancer, we have no choice but usage of this data base. To compensate for the TCGA limitation, we also investigated the relation between CITED2 expression and tumor-free survival rate using IHC of patients' specimens (Fig.1f). The IHC analysis confirmed that patients' survival rate inversely correlates with CITED2 expression.

5) Regarding Figure 2F, it will be important to quantify the data and present them as enrichment as a % of input. It would also be important to show a negative control for ERG IP, i.e. ChIP-qPCR on CITED2 promoter region which is not bound by ERG.

Revision: Fig.2f in the previous manuscript is moved to Fig.2g. As suggested, the ChIP-qPCR data were presented as % of input. Fig.2g includes the P1 and P3 binding of ERG as negative controls that are not bound by ERG. Text revision: line 25 on p5; lines 13 to 17 on p24.

6) The screen for CITED2 interactors has been performed in HEK293T (embryonic kidney cells) – please provide the justification for using this cell type given that the main focus of the paper is the role CITED2 in prostate cancer. It is surprising the majority for further validation experiments are also performed in HEK293T cells.

Revision: We used HEK293T to see the interactions of ectopically expressed proteins because this cell line is the best one for transfection study. In particular, a high transfection efficiency must be achieved to clearly detect PTMs like acetylation and methylation. We believe that the protein-to-protein interactions were not limited to specific cell types. For the interaction among endogenous proteins, IP experiments were performed in prostate cancer cells, as shown in Fig.3b. To further validate our results, *in vitro* IP using recombinant proteins was added to Fig.3d, which supports that the protein interaction is not specific to cell types.

7) IP experiments in Figure 3D have some technical issues. Anti-CITED2 IP in LNCaP cells results in a very weak CITED2 band, in fact it is hardly detectable. Given that CITED2 is expressed at weak levels even in the input, it is difficult to explain why anti-CITED2 IP pools down high levels of its potential interactors. Furthermore, anti-CITED2 IP in PNT2 cells showed no interaction with PRMT5. Please clarify and provide more convincing WBs. :

Revision: Fig.3d in the previous manuscript is moved to Fig.3b. Weak WB data in prostate cancer cell line were replaced with better ones. As PNT2 is not a cancer cell lines, we removed the PNT2 results.

8) Figure 3E shows immunofluorescence images in HEK293T cells. Please show equivalent images in prostate cancer cell lines.

Revision: We newly performed the immunofluorescence assay in prostate cancer cell PC3, which can be found in Fig.3e.

9) HA-p300 WB in Figure 3F seems to show several bands. Please show a positive control for this blot and indicate a specific band and non-specific bands in an image which is not so extensively cropped.

Revision: Western blot was performed again to get clear blots, as shown in Fig.3f.

10) The number of replicate experiments needs to be stated for all IP experiments and molecular weights need to be shown for WBs.

Revision: All IP experiments were done three times, and this statement was added in Figure legend. Molecular weight indications are added to all Western blot bands.

11) Regarding the statements: "PRMT5, and NCL were identified to bind to the transactivation domain (TAD), the serine/glycine-rich junction (SRJ), and the cysteine/arginine-rich domain 3 (CR3) of CITED2, respectively (Fig. 3F, G)" - these are very artificial experiments. It would be essential to generate specific deletion constructs lacking the indicated domains and demonstrate that this abolishes the claimed interactions. At the moment it is difficult to determine whether CITED2 binds all the listed interactors directly or indirectly (i.e. is a part of a large complex with contains P300/CBP, as published previously, and other interactors). :

Revision: To determine if the binding is direct or not, we performed *in vitro* binding assay using recombinant proteins of NCL, CITED2, P300 and PRMT5. The corresponding data is displayed in Fig.3d.

12) The graphic summary shown in Figure 4I suggests that CITED2 is not expressed " without gene fusion " (left panel). However, CITED2 is expressed in HEK293T cells and therefore this graph is unlikely to be correct. Furthermore, most of the IP experiments were performed in HEK293T cells and therefore it is inappropriate at this stage to extend these data to prostate cancer cells. The right panel suggests that CITED2-containing complexes can be found in the cytosol - what is the experimental evidence that CITED2 is present in the cytosol?

Revision: Fig.4i in the previous manuscript is moved to Fig.4k in the revised manuscript. Actually, we did not want to argue that CITED2 is not expressed without gene fusion. We wanted to point out that ERG overexpression due to the gene fusion can contribute to a high expression of CITED2. We revised Fig.4k to avoid reader's confusion. In addition, Leptomycin B (a nuclear export inhibitor) was used to distinguish between cytosolic and nuclear fractions more clearly. When cells were treated with leptomycin B, nuclear exports of CITED2, P300, RioK1, and NCL were inhibited. This suggests that the detected cytosolic complex was due to translocation of the proteins. PRMT5 and WDR77 were found in both nuclear and cytosolic fractions regardless of leptomycin B treatment. The corresponding data can be found in Fig.4a. Text revision: lines 11 to 15 on p7.

13) GSEA shown in Figure 5A cannot be interpreted as indicative of CITED2 functions. Although CITED2 expression may be increased in the interrogated cells, many genes can be overexpressed too. It would have been more informative to knock down CITED2 expression followed by RNA-seq analyses. :

Revision: As suggested, PC3 cells were transfected with si-Control or si-CITED2 and subjected to RNA-Seq analysis. The gene sets identical to those in the previous GSEA analysis were derived. The data can be found in Fig.5c and Fig.S6. Text revision: lines 13 to 17 on p8; lines 22 to 26 on p26.

14) Data shown in Figure 5C are not convincing. No striking differences can be seen between Control and CITED2 cells. The data are not quantified using a larger number of cells; the images show several cells. Major conclusions cannot be drawn from this experiment.

Revision: It is difficult to quantify the change in cytoskeleton arrangement through F-actin staining alone. For this reason, we support our conclusion through additional methods such as migration/invasion assays (Fig.5d and Fig.S7), and EMT marker measurements (Fig.5e). The previous Figure 5C is moved to Figure S7a. Text revision: lines 21 to 24 on p8.

15) The authors conclude that CITED2 may be a therapeutic target for prostate cancer treatment. However, CITED2 inhibition, if this can be pharmacologically achieved, is likely to have severe consequences, e.g. acute bone marrow failure. This has not been discussed at all.

Answer: We agree that complete inhibition of CITED2 will likely bring adverse effects. Instead of completely abolishing CITED2 expression, therapeutic aim could focus on restraining the aberrant overexpression of CITED2. We added this statement in Discussion section. Text revision: lines 1 to 3 on p13.

Reviewer #4 (Expertise: Nucleolin, cancer, Remarks to the Author):

This study claims that: 1) CITED2 is highly expressed in metastatic prostate cancer because of TMPRSS2-ERG gene fusion, which promoted metastasis by activating NCL at the post-translational level, and 2) CITED2-nucleolin axis stimulate cell migration by inducing the epithelial– mesenchymal transition promoting prostate cancer metastasis in mice. This study shows a novel molecular mechanism by which CITED2 regulates the oncogenic activity of NCL. In general, the experimental methods used to test the hypotheses were correctly applied and the results support the most of the main conclusions.

However, this study requires some further evidence to strengthen and validate some conclusions and explain some results before to be accepted for publication

CITED2 is overexpressed in prostate cancer because of TMPRSS2-ERG gene fusion

1) Lines 125-127. The data in the Fig. 2C do not support that CITED2 protein was highly upregulated in prostate cancer cell lines expressing ERG. This data show only a possible correlation between CITED2 and ERG expression. Correct this strong statement “Because the CITED2 protein was highly upregulated in prostate cancer cell lines expressing ERG (Fig. 2C), we further examined the role of ERG in CITED2 expression.” LNCaP, DU145, and 22Rv1 do not harbor TMPRSS2 – ERG fusion. Why were not tested prostate cancer cell line harboring TMPRSS2 – ERG gene fusion? :

Revision: We revised a strong statement “Because the CITED2 protein was highly upregulated in prostate cancer cell lines expressing ERG” to “Immunoblotting analysis in various prostate cancer cell lines showed an apparent correlation between ERG and CITED2 expressions”. In addition, we purchased the VCaP cell line from ATCC and performed cell line screening again. Fig.2c shows that ERG and CITED2 expressions are much higher in VCaP cell line than in others. Text revision: lines 16 to 18 on p5.

2)The data indicate a strongly correlation between tumor harboring TMPRSS2-ERG gene fusion and high expression of CITED2. One of the main claims in this study is that TMPRSS2-ERG gene fusion is a driver to increase the expression of CITED2 in metastatic prostate cancer. These data presented for the authors do not support that TMPRSS2-ERG fusion protein promotes overexpression of CITED2. The authors might use cell lines that harbors TMPRSS2–ERG fusion (such as VCaP) and knockdown the expression of wild-type ERG-type to determine the effect on CITED2 expression in presence or absence of androgen. :

Revision: We purchased VCaP cell line from ATCC and performed key experiments in Fig.2. Cell line screening in Fig.2c now includes VCaP cell line, which confirms that the TMPRSS2-ERG fusion results in elevated expressions of ERG and CITED2. Western blotting of ERG knockdown in Fig.2d, and RT-qPCR in Fig.2e were performed in VCaP. The binding between ERG and CITED2 promoter was also confirmed in ChIP (Fig.2f) and reporter assay (Fig. 2g) using VCaP. Furthermore, to examine the relationship between androgen signaling and CITED2, we used VCaP cell that harbors TMPRSS2-ERG fusion, and treated testosterone to further induce ERG, and studied the change in CITED2 expression. Cell lines without the TMPRSS2-ERG fusion were also treated with testosterone, and ERG and CITED2 levels were inspected. As a result, testosterone induced AR response and increased ERG expression in TMPRSS2-ERG positive VCaP, where CITED2 expression was subsequently increased. Such effects were not observed in TMPRSS2-ERG negative cells (Fig.2f). Text revision: lines 20 to 24 on p5.

CITED2 binds to a multimeric complex consisting of NCL, p300, and PRMT5.

1)The Fig 3d shows endogenous PRMT5, NCL, WDR77, RIOK1, and CITED2 interact to form a multicomplex in PC3, DU145, LNCaP and PNT2 prostate cell lines by immunoprecipitation of CITED2 and analyzing co-immunoprecipitation of the proteins of interest. In the Fig. 2d is showed that these cells do not express (LNCaP and PNT2) or express very low protein levels of CITED2 (PC3 and DU145) compared with 22RV1 and C42B cell lines. How using these negative CITED2- LNCaP and PNT2 cell lines there are the co-immunoprecipitation of proteins of interest? Justify and explain this discrepancy. :

Revision: Fig.3d in the previous manuscript is moved to Fig.3b. We believe you referred to Fig.2c, not Fig.2d, for the cell line screening of CITED2 expression. Western blot in Fig.2c was performed again including VCaP cell line. Although CITED2 and ERG expressions in LNCaP is much weaker compared to VCaP, we can see that LNCaP cell lines are not ‘CITED2-negative’.

2)Supplementary Fig. 4d shows an important finding that suggests that CITED2 might regulate the nuclear translocation of NCL to the cytoplasm. Fig. 3e shows only that all the components of the protein multicomplex

are localized in the nucleus. Fig. 3e should be showed in the supplementary section and Fig. 4d show in paper.

Revision: As suggested, Fig.3e in the previous manuscript was moved to Fig.S4e, and Fig.S4d to Fig.4a.

CITED2 is essential for methylation and acetylation of NCL.

1) Fig. 4a shows clearly that CITED2 overexpression increases the formation of the complex PRMT5/ NCL, but not the complex p300/NCL. Correct the statement or showed convincing data. :

Revision: Fig.4a in the previous manuscript is moved to Fig.4b in the revised manuscript. The experiment was performed again and the old blots were replaced with new ones. As shown, the increase in the p300/NCL complex formation by CITED2 was demonstrated clearly.

2) Are NCL acetylation and methylation required for NCL nucleo-cytoplasmic translocation? :

Answer: Yes! They are. Figures 4b – 4h show that CITED2 promotes the PRMT5-mediated methylation and p300-mediated acetylation of NCL, and Figure 4a shows that CITED2 facilitates the nucleo-cytoplasmic translocation of NCL.

3)Does PRMT5 inhibitor block NCL nucleo-cytoplasmic translocation?

Revision: As suggested, the Fig. 4j experiment was performed using EPZ015666 which is a PRMT5 inhibitor. The nucleo-cytoplasmic translocation of NCL was enhanced by CITED2, which was reversed by PRMT5 inhibition. Text revision: lines 27 to 28 on p7.

The CITED2-NCL axis positively regulates EMT and cell migration in prostate cancer.

1) Fig. 4d. NCL knockdown after 2-4 days results in slow growth and cell death. Indicate the effect of NCL knockdown on cell viability in these cells prior cell migration and invasion assays. Indicate what time after post-transfection of siRNA NCL the cells were tested. :

Revision: We believe you referred to Fig.5d. In PC3 and Du145 with NCL overexpression or knockdown, MTT assays were performed 48 hours after transfection. Cell viability was not significantly affected by NCL knockdown at this time point (Fig.S7f). Text revision: lines 23 to 24 on p8.

The CITED2-NCL axis promotes prostate cancer metastasis in mice.

1) Show that the tumors infected with NCL and CITED2 shRNAs have a significant decrease in expression of these proteins compared to the tumors infected with sh-EGF. :

Revision: Thank you for your good suggestion. We have stored tumor tissues in the liquid nitrogen tank, and homogenized these tissues to check CITED2 and NCL proteins using western blotting. Compared to sh-EGF group, CITED2 and NCL expressions were successfully knocked-down in sh-CITED2 and sh-NCL group, respectively (Fig.S8e). Text revision: lines 6 to 8 on p9.

CITED2-activated NCL promotes the AKT-driven EMT by enhancing translation of AKT mRNA.

1) Line 241-243. No makes sense the statement “Because CITED2 overexpression was attenuated by PRMT5 knockdown (Fig. 7J), CITED2 may have enhanced the NCL–AKT mRNA interaction via PRMT5-mediated” There are not data that show that PRMT5 knockdown decreases the expression of CITED2 or to support that CITED2 may have enhanced the NCL–AKT mRNA interaction via PRMT5-mediated. Please, show data or correct this statement.

Revision: Your comment is correct. We revised this statement in result section. Text revision: lines 6 to 7 on p10.

2)Does NCL post-translation modification by PRMT5 modulate the interaction NCL-AKT mRNA? :

Answer: Yes! Our results in Fig.7j (RNA immunoprecipitation) show that PRMT5 knockdown decreases the binding between NCL and AKT mRNA.

3)Wortmannin and LY29002 are inhibitors of PI3K. AKT is the dominant effector of PI3K signaling and consequently PI3K inhibitors block the AKT activation. PI3K has other important effectors that contribute to cancer progression such as PDK1-mTORC2-SGK and Rac signaling. Therefore, to determine that AKT signaling pathway mediates CITED-induced migration and EMT is required knockdown the expression of AKT.

Revision: Thank you for pointing this out. We conducted additional experiment and confirmed that knockdown of AKT suppresses CITED2-mediated cell migration. Compared to the control, CITED2 overexpression increased migration in PC3 and DU145 cells, which was attenuated by AKT knockdown (Fig.7k). Text revision: lines 9 to 12 on p10.

4)The figure 7k and l are missing important controls: 1) cells over expressing CITED2 without inhibitor treatment 2) cells transfected with siRNA CITED2 without inhibitor treatment. Indicate if the sample labeled without inhibitor, CITED2 and siCITED2, the cells were transfected with the empty expression plasmid and scramble siRNAs.

Revision: Fig.7k and Fig.7l in the previous manuscript are moved to Fig.S11 and S12. We agree to this comment that we should include the data of the CITED2 overexpression and knockdown controls. First of all, please, see Fig. 5d. In this experiment, we already showed that CITED2 overexpression increases in cell migration but its knockdown brings the opposite effect. To avoid the redundant data, we omitted the control data in the following figures.

Others

1) It is required the author provide detail information about the sources and catalog numbers of antibodies used in each specific immunologic assay (WB, IP, immunofluorescence, RNA precipitation, ChIP assays etc.)

Revision: Catalog numbers of all antibodies used in our study are provided in Supplementary Table 3.

2) Discussion is too long

Revision: We eliminated some parts in the discussion section.